# Filovirus infection disrupts epithelial barrier function and ion transport in human iPSC-derived gut organoids

Elizabeth Y. Flores[1,2], Adam J. Hume[2,3◦], Judith Olejnik[2,3◦], Aditya Mithal[1,4◦], Andrew D'Amico[1], MengWei Yang[1], Pushpinder Bawa[1], Feiya Wang[1], Aoife K. O'Connell[2], Anna Tseng[2,3,5], Nicholas A. Crossland[2,3,5], Gustavo Mostoslavsky[1,6*], Elke Mühlberger[2,3*]

1 Center for Regenerative Medicine (CReM), Boston University Chobanian & Avedisian School of Medicine and Boston Medical Center, Boston, Massachusetts, United States of America, 2 National Emerging Infectious Diseases Laboratories (NEIDL), Boston University Chobanian & Avedisian School of Medicine, Boston, Massachusetts, United States of America, 3 Department of Virology, Immunology & Microbiology, Boston University Chobanian & Avedisian School of Medicine, Boston Massachusetts, United States of America, 4 Current address: Department of Medicine, Beth Israel Deaconess Medical Center and Harvard Medical School, Boston, Massachusetts, United States of America, 5 Department of Pathology and Laboratory Medicine, Boston University Chobanian & Avedisian School of Medicine, Boston, Massachusetts, United States of America, 6 Department of Medicine, Section of Gastroenterology; Department of Virology, Immunology, and Microbiology. Boston University Chobanian & Avedisian School of Medicine, Boston Massachusetts, United States of America

◦ These authors contributed equally to this work.
* muehlber@bu.edu (EM); gmostosl@bu.edu (GM)

## Abstract

Gastrointestinal (GI) dysfunction, characterized by severe diarrhea and dehydration, is a central contributor to morbidity and mortality in filovirus disease in patients, yet the role of the epithelium in this clinical outcome remains poorly defined. Here, we employ induced pluripotent stem cell (iPSC)-derived human intestinal (HIOs) and colonic organoids (HCOs) to model Ebola virus (EBOV) and Marburg virus (MARV) infection. These organoids are permissive to filovirus infection and support viral replication. Bulk RNA sequencing revealed distinct intestinal and colonic epithelial responses, including apical and junctional disruption and a delayed virus-specific induction of interferon-stimulated genes. Moreover, infection impaired adenylate cyclase signaling and CFTR-mediated ion transport, providing mechanistic insight into virus-induced secretory diarrhea. This platform recapitulates key features of human GI pathology in filoviral disease and serves as a powerful system to dissect host-pathogen interactions and identify therapeutic targets.

## Author summary

Ebola virus (EBOV) and Marburg virus (MARV) are among the most lethal viruses known. Infection with these viruses leads to severe disease and death.

**Data availability statement:** All relevant data are within the manuscript and its supporting information files. Pluripotent stem cell lines used in this study, along with maintenance standard operating procedures and directed differentiation protocols, are available from the CReM iPSC Repository at Boston University and Boston Medical Center and can be found at (https://crem.bu.edu/cores-protocols/).The RNA-seq data have been deposited in the Gene Expression Omnibus (GEO) under accession numbers GSE298600 and GSE300073. All microscopy image data supporting this study are publicly available through the Zenodo repository. The dataset includes .tif files (some merged and some individual fluorescence channels used in the published figure panels), .czi files (multi-channel and z-stack raw microscopy data), and .gif files representing animated z-stack projections in support of the data shown in Fig 3 ( https://doi.org/10.5281/zenodo.17587777; https://zenodo.org/records/17587777), S1 Fig (https://doi.org/10.5281/zenodo.17586960; https://zenodo.org/records/17586960), and Fig 6 (https://doi.org/10.5281/zenodo.17586449; https://zenodo.org/records/17586449). Additional histopathology images in support of Fig 3, S3 and S5 Figs are accessible through the Zenodo repository under https://doi.org/10.5281/zenodo.17583211 (https://zenodo.org/records/17583211).

**Funding:** This work was supported by the National Institutes of Health (NIH) National Institute of Allergy and Infectious Diseases (R21AI167369 to GM and EM; https://www.niaid.nih.gov), and by the Howard Hughes Medical Institute Emerging Pathogens Initiative (Agreement dated 9/16/22; Lead Investigator Anna Pyle; GM and EM; https://www.hhmi.org). Histopathology analyses were supported by NIH S10 instrumentation awards (S10OD030269 and S10OD026983). EYF was supported by the Boston University Clinical and Translational Science Institute (BU CTSI) TL1 Predoctoral Fellowship in Regenerative Medicine (TL1TR001410 to EYF) and an NIH NIAID Predoctoral Fellowship (F31AI183803 to EYF). The funders had no role in study design, data collection and analysis, decision to publish, or preparation of the manuscript.

**Competing interests:** The authors have declared that no competing interests exist.

One of their most harmful effects is damage to the gastrointestinal tract, causing intense diarrhea and life-threatening dehydration. Yet, how these viruses affect the gut remains poorly understood. In this study, we used human mini-guts—small, three-dimensional tissues grown from stem cells that mimic the human intestinal and colonic epithelium—to investigate how these viruses interact with gut epithelial cells. We found that both EBOV and MARV infect and replicate in these tissues, disrupt key barrier structures, and interfere with the cells' ability to regulate fluid secretion. These effects mirror the severe symptoms seen in patients. Our study provides new insight into how EBOV and MARV damage the gut and identifies specific cellular pathways that may be targeted for treatment. This research not only improves our understanding of EBOV and MARV infections but also offers new infection platforms for testing therapies aimed at protecting the gastrointestinal system during filovirus outbreaks.

## Introduction

Ebola virus (EBOV) and Marburg virus (MARV) are members of the filovirus family and have caused multiple devastating outbreaks since their identification in 1976 and 1967, respectively [1–3]. With case fatality rates remaining alarmingly high, filoviral infections continue to pose a significant risk for widespread epidemics, underscoring the urgent need for comprehensive strategies to combat these lethal pathogens [4]. Gastrointestinal (GI) complications, including severe diarrhea, dehydration, and fluid loss, are hallmark symptoms of filovirus disease and play a central role in the fatal outcome of EBOV and MARV infections, contributing to hypovolemic and septic shock [3,5–8]. While animal models, particularly non-human primates (NHPs), have provided valuable insights into the mechanisms of filovirus disease, they fail to fully recapitulate the intricate GI pathology seen in human cases [9,10]. Filling in this critical gap in our understanding of the molecular mechanisms of filoviral-induced GI dysfunction may provide insights into the transmission and pathogenesis of filoviruses and could aid in the development of effective filovirus disease treatments.

Given the limitations of NHP models and the challenges associated with human tissue availability, more accurate, human-based in vitro models are highly desirable. Recent innovations in the development of human intestinal organoids (HIOs) derived from induced pluripotent stem cells (iPSCs) present a promising alternative [11–18]. These organoid models closely recapitulate the architecture and function of the human GI epithelium, allowing studies of virus-host interactions at the cellular and molecular levels. Our iPSC-derived model is unique in its ability to differentiate into both proximal small intestine and distal colonic lineages, enabling a comprehensive assessment of viral impacts across distinct regions of the gut [19].

In this study, we employed iPSC-derived HIOs and human colonic organoids (HCOs) as region-specific models to investigate the pathogenesis of filoviral infection. These organoid systems recapitulate the structural and functional attributes of the human intestinal epithelium and provide a relevant and reproducible platform for

studying viral-host interactions. We demonstrate that both HIOs and HCOs are permissive to EBOV and MARV infection with robust viral replication, enabling comprehensive transcriptomic profiling of the epithelial-intrinsic host response. Our data reveal that infection with EBOV and MARV induces a rapid and substantial transcriptional reprogramming, characterized by the upregulation of pro-inflammatory signaling cascades, epithelial-to-mesenchymal transition (EMT) markers, hypoxia-associated genes, and metabolic pathways. MARV infection elicited a delayed interferon-stimulated gene (ISG) response, a feature not observed in EBOV-infected organoids. Additionally, infections with both viruses led to the significant dysregulation of genes involved in apical surface structure and tight junction integrity, correlating with increased viral replication and morphological changes indicative of epithelial disruption.

Forskolin-induced swelling assay revealed region-specific responses to viral infection, highlighting differential susceptibility along the intestinal axis which closely mirrors clinical manifestations such as secretory diarrhea, frequently documented in filovirus-infected individuals [20].

Together, these findings establish iPSC-derived HIOs and HCOs as a robust and scalable model system to study filovirus infection in the human intestinal epithelium. This system offers a powerful platform for investigating the molecular mechanisms of filoviral intestinal pathology and may provide an invaluable tool for preclinical evaluation of therapeutic interventions aimed at preserving epithelial integrity during viral infection.

## Results

### iPSC-derived human intestinal organoids (HIOs) and colonic organoids (HCOs) are permissive to EBOV and MARV infection and replication

To model region-specific gastrointestinal infection, we generated human iPSC-derived intestinal (HIO) and colonic (HCO) organoids using a modified protocol [17], yielding proximal (small intestine–like) and distal (colon-like) epithelium, respectively, in the absence of mesenchyme (Fig 1A–1E). Single-cell RNA sequencing confirmed robust expression of intestinal markers including *CDX2* and *Villin1*, with multicellular complexity reflective of native human gut epithelium [19].

We employed this model to assess the susceptibility of HIOs and HCOs to EBOV infection, using recombinant EBOV expressing ZsGreen as a reporter gene [21]. HIOs and HCOs were derived from two genetically distinct iPSC donors, with three or more independent differentiations per donor line. HIOs and HCOs were infected with EBOV-ZsGreen at a multiplicity of infection (MOI) of 10 on day 35 of culture. At 1 day post infection (dpi), robust EBOV-ZsGreen expression was observed in all organoids, characterized by numerous ZsGreen-positive columnar basal cells distributed throughout the epithelial monolayer (Fig 2). In contrast, mock-infected organoids showed no ZsGreen-positive cells under identical fluorescent exposure conditions. Viral spread became evident by 3 dpi, with a noticeable increase in the number of infected cells across the organoid cultures (Fig 2). By 7 dpi, organoids exhibited significant cytopathic effects (CPE), leading to the eventual collapse of the organoid structures (Fig 2). Due to the extent of virus-induced damage, organoids at 7 dpi were not included in downstream analyses. These results establish that both HIOs and HCOs are permissive to EBOV infection, supporting viral replication and virus-induced cell damage in a regionally defined human gut model.

Building on our work with the fluorescent EBOV-ZsGreen virus model and following the determination of optimal infection timepoints, we next investigated the susceptibility of HCOs to wild-type EBOV (Mayinga isolate) and MARV (Musoke isolate) at an MOI of 10. Confocal microscopy was employed to visualize caudal type homeobox 2 (CDX2)-GFP labeling, marking intestinal identity in green, and red fluorescence corresponding to immunofluorescence staining of filoviral nucleoprotein (NP) or nucleocapsid (NC), which confirmed viral infection. At 1 dpi, minimal viral presence was detected, characterized by small, punctate fluorescence signals for both EBOV and MARV (Fig 3A and 3E). By 3 dpi, viral expression was widespread throughout the organoid culture, indicating active viral replication and dissemination (Fig 3B and 3F). Higher magnification imaging revealed the presence of inclusion bodies, a hallmark of filovirus infection, localized within the perinuclear region of infected cells (Fig 3C and 3G). These findings confirm that iPSC-derived distal HCOs are permissive to

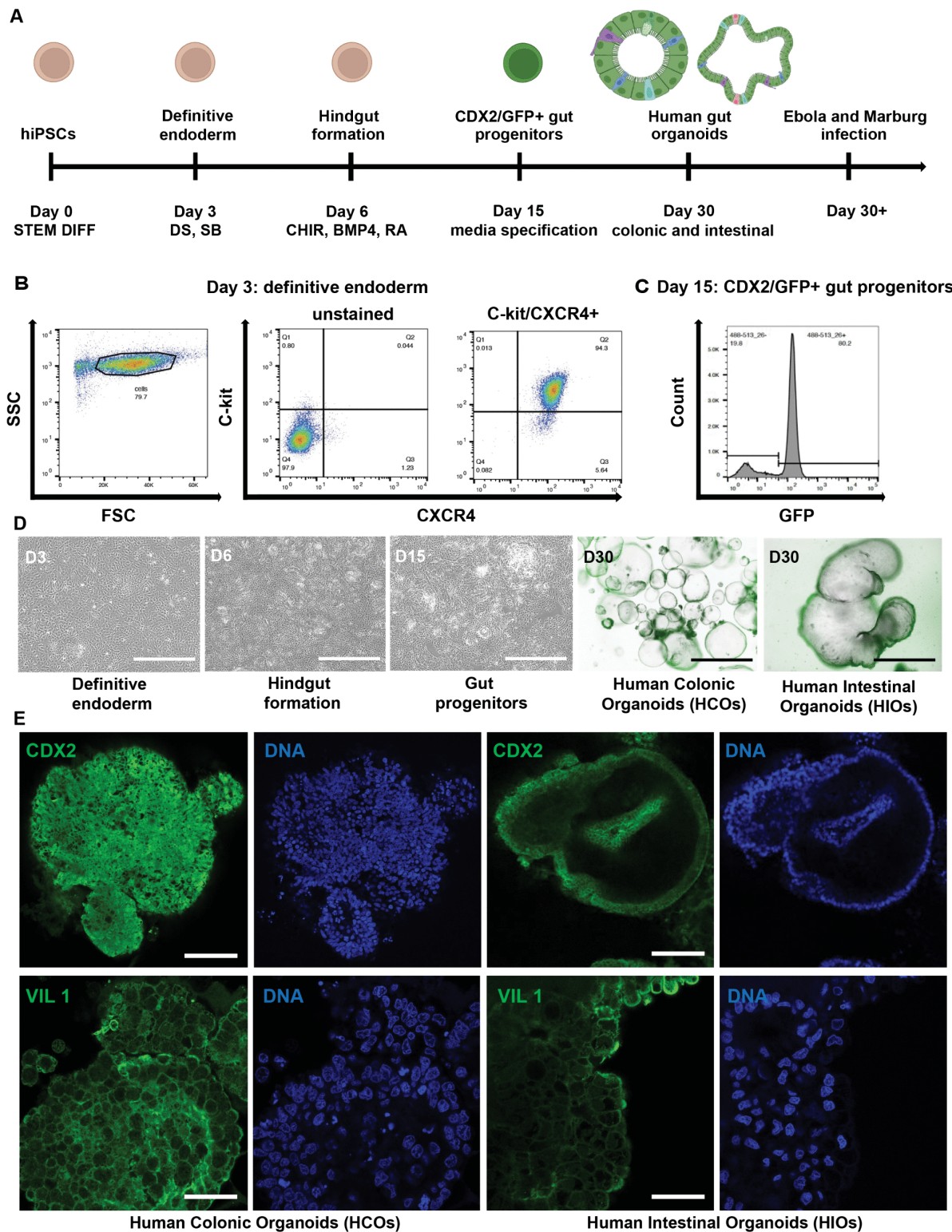

**Fig 1. Directed differentiation of human iPSCs into region-specific intestinal (HIOs) and colonic (HCOs) organoids. (A)** Schematic representation of the stepwise differentiation protocol used to generate gut organoids from human induced pluripotent stem cells (iPSCs). Illustrations created in BioRender. Muhlberger, E. (2025) https://BioRender.com/14baw98. **(B)** BU1 CDX2-GFP iPSCs were differentiated to definitive endoderm by day 3. **(C)** A CDX2-eGFP knock-in reporter line was used to monitor the emergence of hindgut progenitors during differentiation. **(D)** Gut progenitor cells were

further specified into regional identities using defined media: colonic organoids (HCOs) were cultured in CKDCI media, and intestinal organoids (HIOs) were cultured in media containing CHIR, KGF, EGF, R-spondin, and Noggin. Brightfield and fluorescence imaging was performed at day 30 to visualize organoid morphology. Images were captured using a Keyence BZ-X710 fluorescence microscope. **(E)** Immunofluorescence staining was used to detect CDX2 and villin (VIL) (green) and Hoechst for nuclear staining (blue) in HIOs and HCOs. Confocal imaging was performed using a Zeiss LSM 710-Live Duo Confocal microscope with two-photon capability. Scale bars = 100 μm. Data shown are representative of n = 3 independent differentiations.

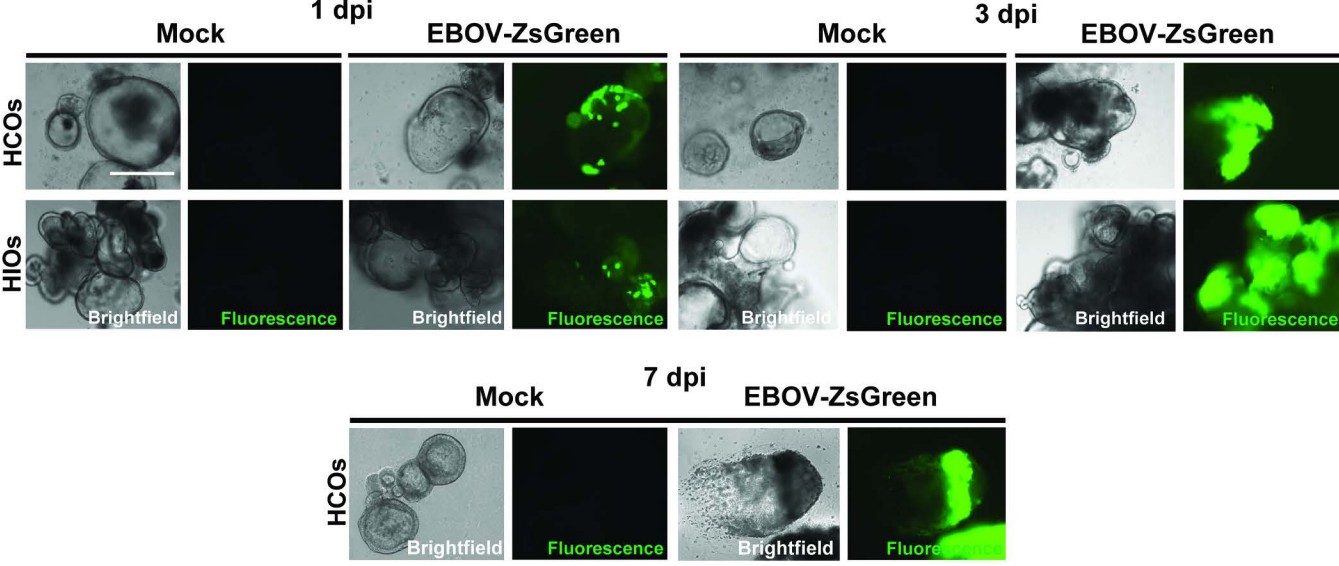

**Fig 2. iPSC-derived HIOs and HCOs are permissive to EBOV infection.** HIOs and HCOs were infected with EBOV expressing ZsGreen (EBOV-ZsGreen) at an MOI of 10 on day 35 of differentiation. Brightfield and fluorescence images of live organoids were acquired at 1, 3, and 7 dpi with a fixed exposure time of 25 ms. Imaging was performed using the EVOS M50000 Imaging System. Scale bar = 100 μm.

both EBOV and MARV infection. Additionally, comparable infection efficiencies were observed in iPSC-derived HCOs from a genetically distinct donor, further supporting the robustness of this model (S1 Fig). The BU310-Cre2 iPSC line, used in this experiment, lacks the CDX2-GFP knock-in promoter. Consequently, HCOs were sorted based on CD26 (dipeptidyl peptidase 4 (DPP-4)) positivity, a marker of intestine identity and a potential biomarker for intestinal health [22]. To further confirm intestinal identity, the HCOs were stained for Villin1, an apical brush border protein specific to the intestinal epithelium (S1 Fig).

Flow cytometry analysis, utilizing an antibody against EBOV NP, further quantified the infection rate, revealing that approximately 42% of cells were infected with EBOV by 3 dpi (Fig 3D). Attempts to quantify MARV infection using flow cytometry were unsuccessful, likely due to antibody incompatibility and the stringent viral inactivation protocols required for BSL-4 pathogen handling. Therefore, immunohistochemistry (IHC) was employed to quantify both EBOV and MARV infections, providing spatially resolved and reproducible measures of viral spread within the organoids. These findings were corroborated by IHC, where the localization and expression patterns of EBOV matrix protein VP40 and MARV glycoprotein (GP) mirrored those observed by immunofluorescence assay. Infection progression was characterized by minimal viral presence at 1 dpi, which increased significantly by 3 dpi, in contrast to mock-infected controls (Fig 3H and 3J). This time-dependent progression in viral replication and spread was supported by quantifying viral antigen positive tissue areas, with similar trends observed for both EBOV and MARV (Fig 3I and 3K). These results confirm that HCOs are permissive to both EBOV and MARV infections, facilitating efficient viral replication, dissemination, and the induction

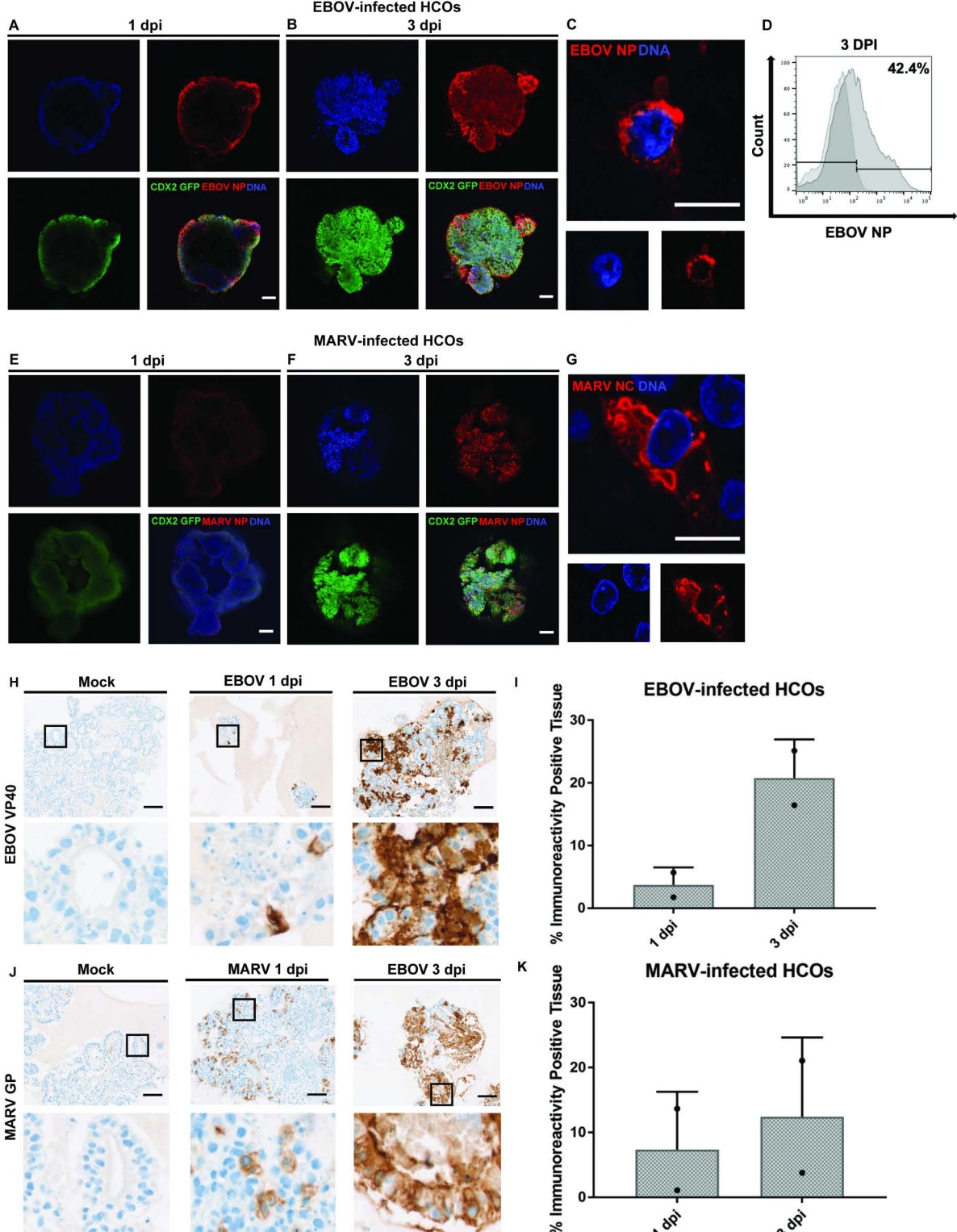

**Fig 3. iPSC-derived HCOs are permissive to EBOV and MARV infection.** Human iPSC-derived colonic organoids (HCOs) were infected on day 35 of differentiation with EBOV or MARV at an MOI of 10. **(A-C, EBOV infection; E-G, MARV infection)** Viral infection was evaluated at 1 and 3 dpi using confocal microscopy. Immunofluorescence staining was performed with antibodies against viral nucleoproteins (NP for EBOV and NC for MARV; red),

CDX2-GFP to mark colonic epithelium (green), and Hoechst for nuclear staining (blue). Images were acquired using a Zeiss LSM 710 Live-Duo confocal microscope with two-photon capability. Panels A, B, E, F: scale bars = 100 μm; panels C, G: scale bars = 10 μm. IF data are representative of three independent experiments (n = 3). **(D)** Flow cytometry was used to quantify EBOV infection using NP-specific antibodies in conjunction with a viability dye to assess infected, live cells. **(H, J)** Immunohistochemistry was performed to detect EBOV VP40 (H) and MARV GP (J) antigens in fixed organoid sections. Insets show higher-magnification views used for quantification of viral spread and antigen intensity. **(I, K)** Quantification of immune-reactive tissue. IHC data are representative of two independent infections (n = 2).

of cytopathic effects. Together, the increase in antigen-positive cells, broader spatial distribution of viral RNA, and rising viral transcript levels support intra-organoid viral spread, although direct measurement of infectious virus release into the supernatant was not performed.

### RNA sequencing of infected iPSC-derived HCOs and HIOs reveals an epithelial-intrinsic apical and junctional disruption and delayed interferon stimulated ene Responses

Having established a human colonic epithelial model for EBOV and MARV infection using iPSC-derived HCOs, we next aimed to delineate the global, time-dependent transcriptomic responses of these organoids to filoviral infection. To investigate the host epithelial responses to filovirus infection, we performed bulk RNA sequencing (RNA-seq) on HCOs infected with EBOV or MARV (Fig 4A). Organoids were infected at day 35 of differentiation and harvested at 1 and 3 dpi. Transcriptomic profiles were compared to mock-infected controls (n = 3 biological replicates per condition). Principal component analysis (PCA) revealed time-dependent transcriptional changes, with the largest variation attributed to post-infection time (PC1, 19.3%) and infection status (PC2, 16.9%) (Fig 4B and 4C). By 3 dpi, infected organoids exhibited distinct separation from controls, reflecting progressive viral replication and host response. LOESS regression confirmed increasing viral RNA expression in EBOV- and MARV-infected samples over time, with no viral transcripts detected in mock-infected controls (S2 Fig). Read counts of viral transcripts confirmed productive viral replication in both infection models (Fig 4D and 4E).

To characterize virus-specific transcriptional signatures, we identified differentially expressed genes (DEGs) relative to mock-infected controls (LogFC > 2 or < –2; p < 0.05). Venn analysis revealed both shared and virus-specific DEGs (Fig 4F). At 1 dpi, 160 upregulated genes were common to both EBOV and MARV, while no significantly downregulated genes were shared. By 3 dpi, 106 upregulated and 19 downregulated genes overlapped between the two infections, highlighting distinct and evolving host responses.

At 1 dpi, both viruses induced significant enrichment of pathways associated with hypoxia, EMT, TNF-α signaling, inflammatory responses, glycolysis, bile acid metabolism, and key epithelial programs, including apical surface and tight junction integrity (Fig 4G). These pathways are essential for maintaining intestinal epithelial polarity, selective permeability, and absorptive function. Their dysregulation suggests an early disruption of epithelial architecture and barrier function, which may underlie clinical features such as malabsorption and diarrhea commonly observed in filovirus-infected patients. Simultaneous downregulation of DNA repair mechanisms further indicates compromised epithelial cell homeostasis and increased vulnerability. By 3 dpi, many of these transcriptional signatures persisted; however, a key divergence emerged in interferon signaling (Fig 4H). MARV-infected HCOs exhibited sustained and significant upregulation of interferon response pathways, whereas EBOV-infected HCOs demonstrated suppression of these same pathways. This differential regulation highlights a virus-specific modulation of the host innate immune response during the later stages of infection.

Unsupervised hierarchical clustering of the top 250 DEGs revealed virus- and time-specific transcriptional profiles (Fig 5A). As expected, viral transcripts were among the most highly upregulated genes, confirming active replication. In addition to viral gene expression, infection induced significant transcriptional changes in host gene sets associated with critical intestinal epithelial functions. These included markers of hypoxic stress, glycolytic metabolism, EMT, apical membrane organization, tight and gap junction integrity, and core regulators of absorptive and barrier function. By 3 dpi,

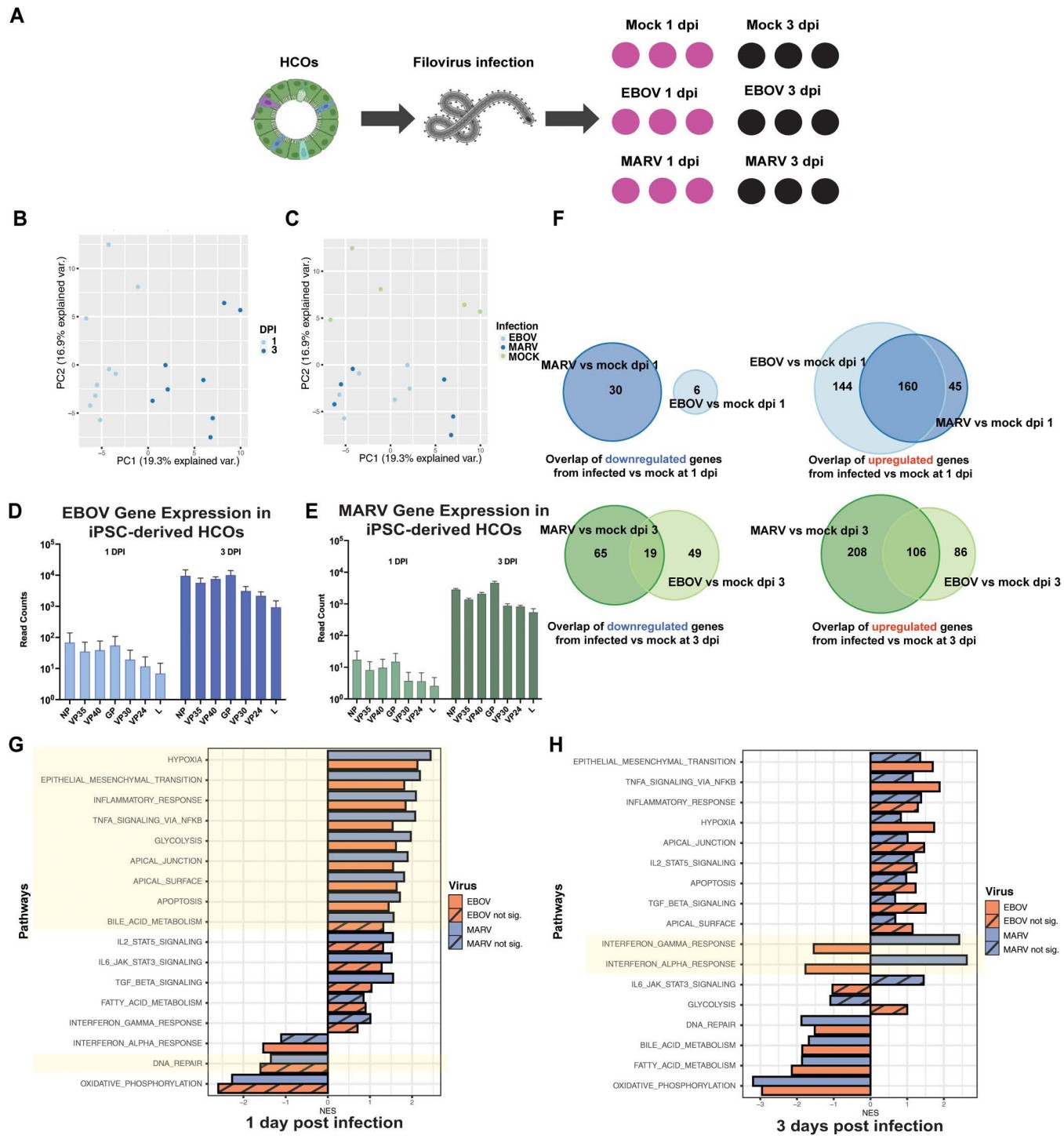

**Fig 4. Transcriptomic profiling of EBOV- and MARV-infected HCOs reveals viral load and shared and unique host responses. (A)** Schematic overview of the bulk RNA-seq experimental workflow in iPSC-derived proximal human intestinal organoids (HIOs). Illustrations created in BioRender. Muhlberger, E. (2025) https://BioRender.com/nncviy6. **(B and C)** Principal component analysis (PCA) of the distal iPSC-derived HCO transcriptomic response to EBOV and MARV infections at 1 and 3 dpi. **(D)** Read counts of EBOV transcripts at 1 (light blue) and 3 (dark blue) dpi. **(E)** Read counts of MARV transcripts at 1 (light green) and 3 (dark green) dpi. **(F)** Venn diagrams illustrating the number of common and unique significantly differentially expressed genes (DEGs) in MARV and EBOV-infected iPSC-derived HCOs at different infection time points. Genes that were upregulated (LogFC>2, p<0.05) and downregulated (LogFC<-2, p<0.05) at each time point (1 and 3 dpi) are shown. Gene set enrichment analysis (GSEA) was performed

using the Hallmark gene sets on the top DEG sets in distal HCOs at **(G)** 1 dpi and **(H)** 3 dpi, compared to mock-infected controls, for both EBOV and MARV infections. Striped bars indicate non-significant results, while solid bars represent statistically significant findings (p < 0.05).

MARV-infected HCOs showed robust induction of ISGs, including *OASL*, *MX1, IFIT1, IFIT2, IFI6*, and *CXCL10* (Fig 5B), consistent with a strong innate immune activation. In contrast, EBOV-infected organoids did not exhibit a delayed ISG response.

Transcriptomic data also indicated metabolic and structural disruption. Genes associated with epithelial barrier integrity, including solute carriers (*SLC10A2, SLC9A3, SLC26A3, SLC5A1*) and epithelial adhesion (e.g., *EPCAM*), were significantly downregulated in both infections by 1 dpi (Fig 5C). Additionally, transcripts implicated in diarrheal pathogenesis (e.g., *TTC37, TTC7A, GUCY2C*) were dysregulated. These findings suggest compromised absorptive function and epithelial barrier integrity early in infection, which may underlie gastrointestinal symptoms commonly observed in filovirus disease.

Hallmark EMT-related genes were significantly upregulated by 1 dpi, reflecting the early onset of epithelial remodeling. EMT is characterized by the loss of epithelial polarity and intercellular adhesion, accompanied by a transition to a mesenchymal state with enhanced motility and invasive capacity. This phenotypic shift compromises epithelial barrier integrity, a critical feature of intestinal homeostasis, and may facilitate viral dissemination within the tissue microenvironment. The resulting disruption of barrier function likely contributes to clinical manifestations such as diarrhea and malabsorption, which are common in filovirus-infected individuals. Moreover, EMT activation is associated with the induction of pro-inflammatory signaling cascades and has been implicated in viral immune evasion through suppression of antiviral responses [23].

In parallel, we observed significant dysregulation of oxidative stress response pathways and a pronounced downregulation of oxidative phosphorylation (OXPHOS), particularly at 3 dpi and most prominently in MARV-infected HCOs (Fig 4G). OXPHOS is essential for ATP generation and mitochondrial function in intestinal epithelial cells; its impairment is associated with increased production of reactive oxygen species (ROS), mitochondrial dysfunction, and cellular injury. These changes reflect a virus-induced metabolic shift from OXPHOS to glycolysis, a well-documented strategy exploited by many viruses to support replication by increasing the supply of metabolic intermediates and energy [24,25].

Concurrently, genes associated with hypoxic stress, including *PDK1*, *BNIP3L*, *VEGF*, and *LOX,* were significantly upregulated in both EBOV- and MARV-infected HCOs, consistent with prior reports of hypoxia-related signaling during viral infection [26,27]. This hypoxia gene signature overlapped with EMT activation, suggesting a coordinated cellular stress response involving inflammation, metabolic reprogramming, and epithelial remodeling [28,29] (Fig 5D). Together, these findings suggest that filovirus infections perturb epithelial structure and function, as supported by both transcriptional signatures and immunofluorescence evidence of disrupted epithelial organization (Figs 2 and 3).

To further investigate filovirus infection in distinct regional gut areas, we performed parallel infections using iPSC-derived HIOs (small intestine-like) with EBOV and MARV at an MOI of 10. Confocal microscopy confirmed the intestinal epithelial identity of the HIOs via expression of CDX2-GFP (green), while filoviral nucleoprotein (NP) was detected in infected cells (red), and nuclei were visualized by DNA staining (blue). At 1 dpi, NP signal for both viruses appeared as sparse, punctate foci, indicating limited early infection (Fig 6A and 6D). By 3 dpi, viral expression became widespread, indicative of active replication and dissemination (Fig 6B and 6E), with high-resolution imaging revealing viral inclusion bodies in the cytoplasm of the infected cells (Fig 6C and 6F).

Building on previous findings from HCOs, where virus-specific transcriptional responses were observed at later stages of infection, we performed bulk RNA sequencing of HIOs at 3 dpi in triplicate to characterize the transcriptional landscape and identify viral-specific differences (Fig 7A). Our data show that both EBOV and MARV infect HIOs and produce infectious virions, similar to HCOs. PCA revealed significant, time-dependent shifts in the transcriptional profiles of the infected organoids, with the primary sources of variation attributed to time post-infection (PC1, 35.6%) and viral infection status

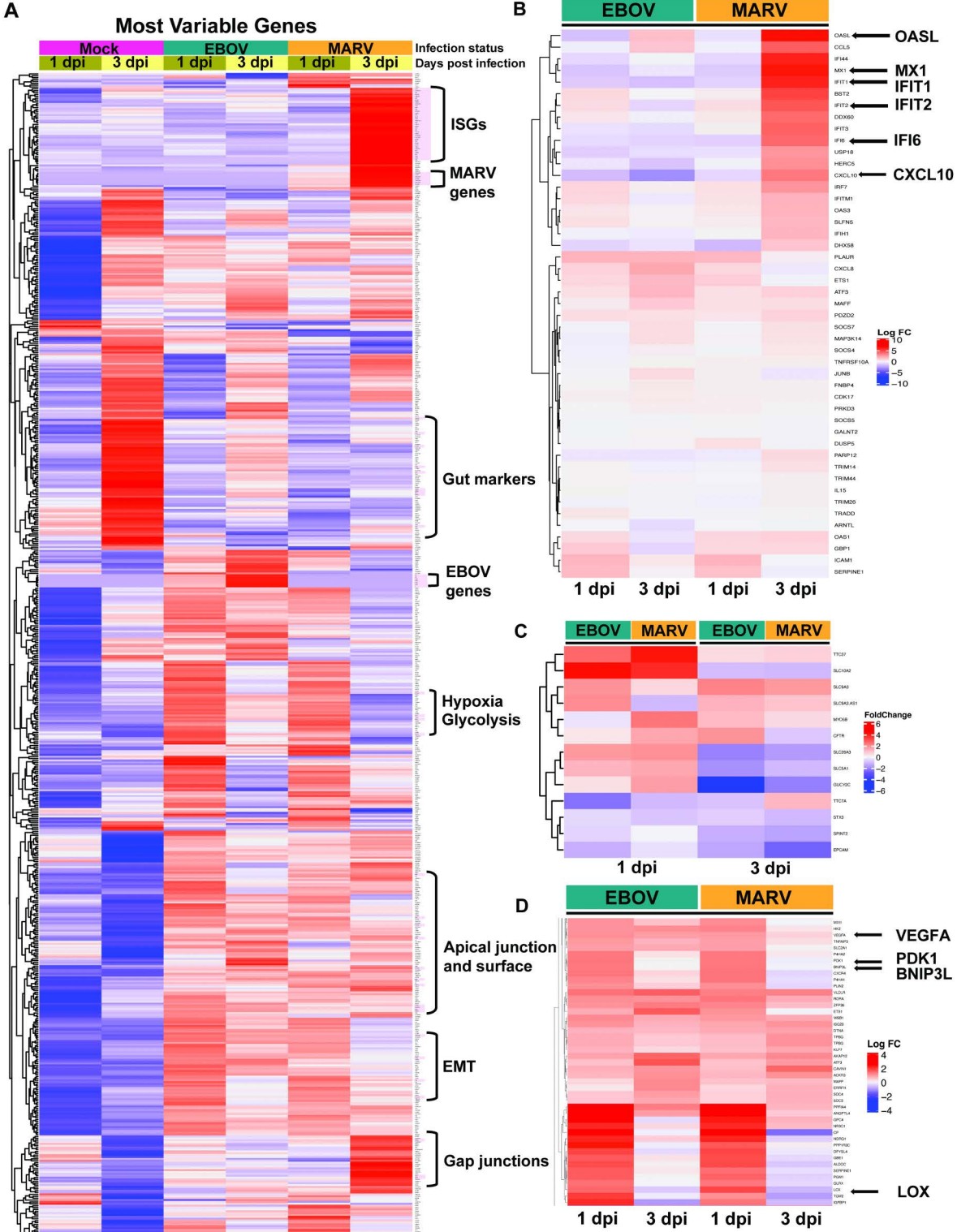

**Fig 5. Delayed ISG induction in MARV infection and dysregulation of diarrhea-associated genes in EBOV- and MARV-infected human colonic organoids. (A)** Heatmap displaying the 250 most variable genes across all time points in EBOV- and MARV-infected HCOs. Differentially expressed genes (DEGs) with the most significant upregulation and downregulation at 1 and 3 dpi are highlighted in red. **(B)** Heatmap illustrating logFC expression values of interferon-stimulated genes (ISGs) in EBOV- and MARV-infected distal HIOs at 1 and 3 dpi. **(C)** Heatmap showing logFC expression values for

a curated panel of genes implicated in diarrheal pathogenesis, measured in HCOs following EBOV and MARV infection at 1 and 3 dpi. **(D)** Heatmap representing logFC expression levels of genes involved in hypoxia responses in EBOV- and MARV-infected distal HIOs at 1 and 3 dpi. Key logFC changes in gene expression are annotated on the right of each heatmap.

(PC2, 20.0%) (Fig 7B). Read counts of viral RNA confirmed productive infection in both EBOV- and MARV-infected samples (Fig 7C and 7D).

Transcriptomic profiling of infected HIOs at 3 days post-infection (dpi) revealed significant upregulation of immune-related gene programs. Both EBOV- and MARV-infected HIOs showed enrichment of type I and III interferon signaling pathways, inflammatory response genes, apoptotic pathways, and components of the JAK/STAT signaling cascade (Fig 7E). Notably, the interferon-related gene expression signature was consistent with patterns previously observed in non-infected bystander cells in EBOV infection [30–32].

Combined In situ hybridization (ISH) and IHC at 3 dpi revealed virus- and region-specific epithelial immune responses and viral antigen respectively, which partially aligned with transcriptomic profiles. Consistent with RNA-seq data, MARV infection elicited significantly greater epithelial expression of *MX1*, *IFNb*, and *CXCL10* mRNA in distal HCOs compared to EBOV infection, demonstrating robust ISG induction in MARV-infected HCOs (S3A–S3C Fig). Overall, we observed little overlap between *MX1*, *IFNb* and *CXCL10* mRNA staining (yellow) and the MARV staining (red), suggesting that mainly non-infected bystander cells were activated by MARV virus infection. In contrast, EBOV-infected HCOs exhibited minimal expression of these markers, indicating limited immune activation. In proximal HIOs, transcriptomic analysis showed upregulation of interferon signaling pathways for both viruses; however, the pronounced MARV-associated ISG induction observed in HCOs was not evident. Nonetheless, ISH revealed increased *MX1*, *IFNb*, and *CXCL10* mRNA levels in MARV-infected HIOs compared to EBOV, indicating comparatively stronger immune activation (S3A–S3C Fig). Notably, *MX1* gene expression was consistently greater in distal HCOs than proximal HIOs across infections, suggesting regional differences in epithelial antiviral responsiveness (S3D–S3F Fig). Collectively, these findings highlight virus- and region-specific innate immune activation, with MARV eliciting a more robust epithelial antiviral response, particularly in non-infected cells in the distal gut.

## Region-specific disruptions in cAMP-mediated ion transport and epithelial function in filoviral-infected HIOs and HCOs

To evaluate the functional consequences of filovirus infection on epithelial ion transport, forskolin-induced swelling assays were conducted in iPSC-derived HIOs and HCOs. This assay provides a quantitative measure of cystic fibrosis transmembrane conductance regulator (CFTR)-mediated fluid transport, which is highly active in the gut and is based on forskolin-driven cAMP signaling and activation of epithelial chloride channels [33].

The forskolin-induced swelling assay is utilized to assess cystic fibrosis transmembrane conductance regulator (CFTR) function by leveraging forskolin's mechanism of action. Forskolin activates adenylate cyclase, which increases intracellular cAMP levels. Elevated cAMP activates protein kinase A (PKA), resulting in phosphorylation of CFTR and subsequent opening of the CFTR chloride channel. This enables chloride and sodium ions to cross the membrane, establishing an osmotic gradient that promotes water uptake and causes organoid swelling. Thus, this assay provides a functional readout of CFTR activity and serves as a tool to assess the impact of CFTR mutations on ion transport and epithelial function.

Treatment with 5 μM forskolin led to significant organoid swelling in mock-infected, EBOV-infected, EBOV-ZsGreen-infected, and MARV-infected HIOs. Swelling was detectable at 24 hours post-treatment and continued through 48 hours, indicating preservation of functional cAMP-dependent ion and fluid transport in the proximal intestinal epithelium (Fig 8A). No swelling was observed in untreated control organoids over the same time course. EBOV-ZsGreen-infected HIOs showed an inverse correlation between viral load and swelling, organoids with lower fluorescence intensity—indicative of lower infection—exhibited greater forskolin-induced swelling, whereas those with higher viral burden displayed attenuated

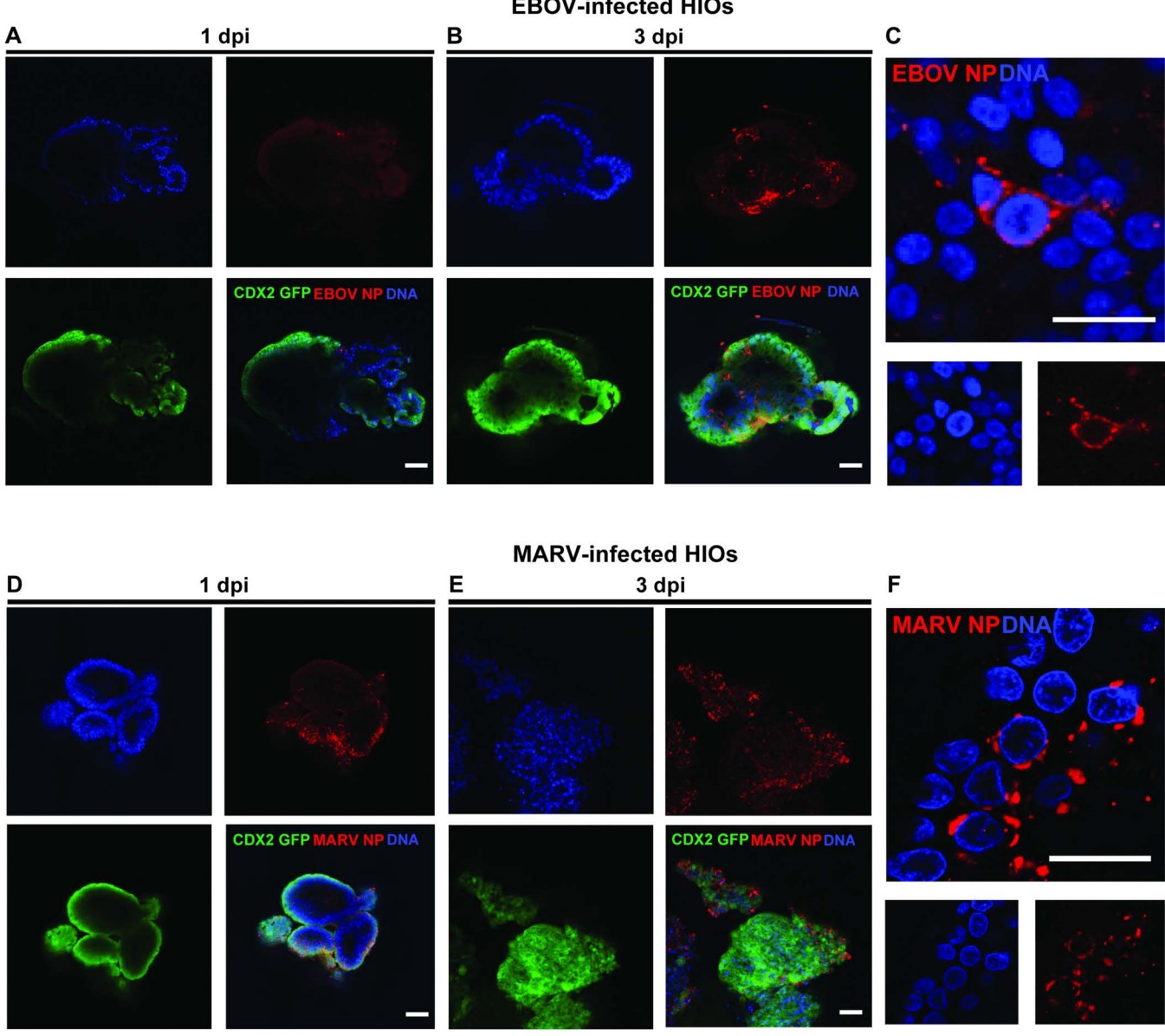

**Fig 6. iPSC-derived HIOs are permissive to EBOV and MARV infection.** Human iPSC-derived intestinal organoids (HIOs) were infected on day 35 of differentiation with EBOV or MARV at an MOI of 10. **(A–C, EBOV infection; D–F, MARV infection)** Viral replication was assessed at 1 and 3 dpi using confocal immunofluorescence microscopy. Staining was performed using antibodies against viral nucleoproteins (NP for EBOV and NC for MARV; red), CDX2-GFP to label intestinal epithelium (green), and Hoechst for nuclear counterstaining (blue). Images were acquired using a Zeiss LSM 710 Live-Duo confocal microscope with two-photon capability. Panels A, B, D, E: scale bars = 100 μm; panels C, F: scale bars = 10 μm. Data are representative of three independent infection experiments (n = 3).

swelling (Fig 8A). MARV-infected HIOs also retained the ability to swell in response to forskolin stimulation over 48 hours, consistent with functional CFTR activity.

Unexpectedly, EBOV- and MARV-infected HCOs exhibited a complete absence of forskolin-induced swelling, with organoid size remaining comparable to untreated controls throughout the 48-hour assay period (Fig 8B). In contrast,

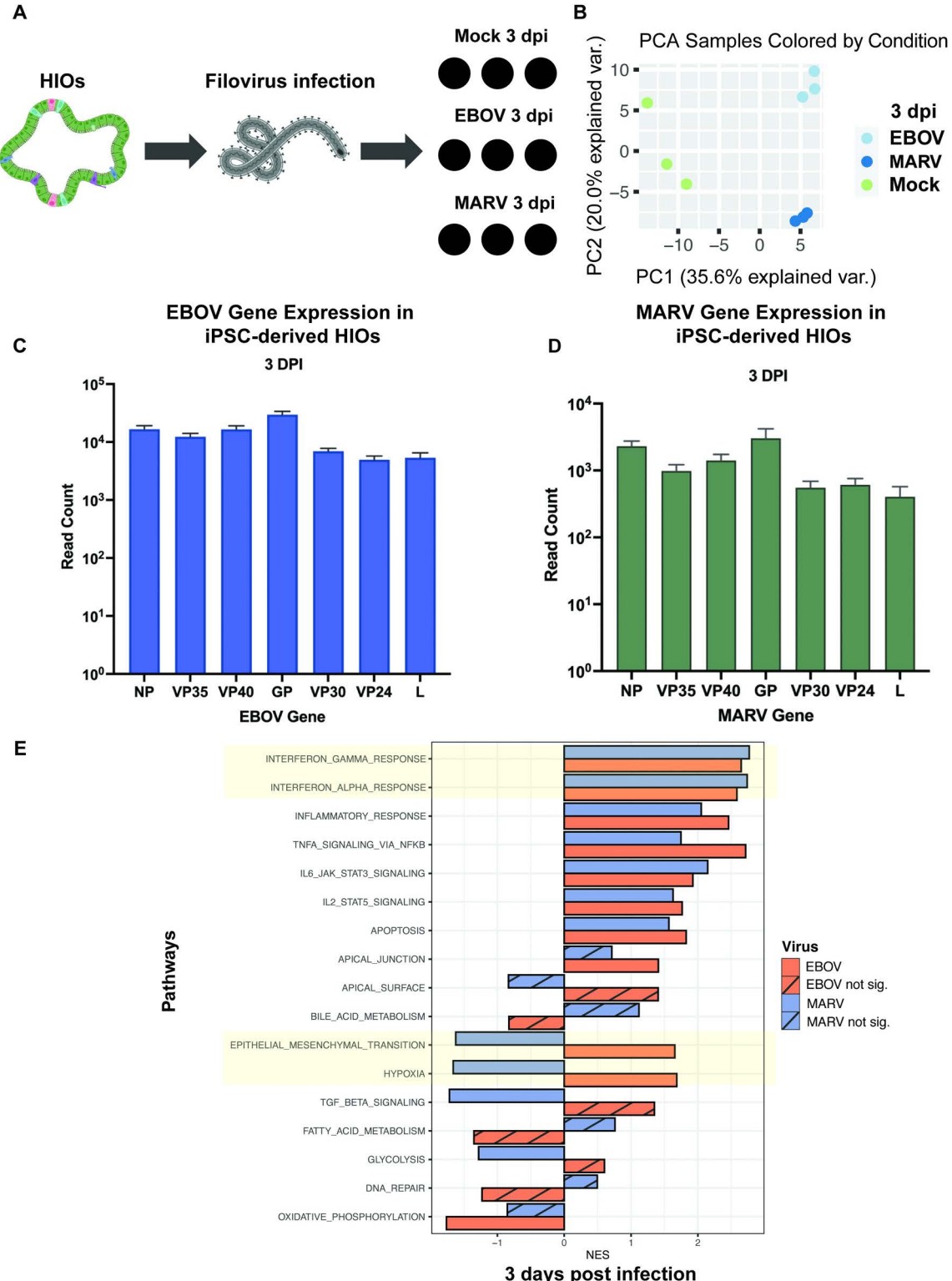

**Fig 7. Transcriptomic profiling of HIOs reveals viral replication and distinct host responses to EBOV and MARV infection. (A)** Schematic overview of the bulk RNA-seq experimental workflow in iPSC-derived proximal human intestinal organoids (HIOs). Illustrations created in BioRender. Muhlberger, E. (2025) https://BioRender.com/fwnhzjr. **(B)** Principal component analysis (PCA) illustrating global transcriptomic variation in proximal HIOs following infection with EBOV and MARV. **(C–D)** Quantification of viral transcript abundance at 3 dpi for EBOV (dark blue) and MARV (dark green),

indicating active replication in proximal HIOs. **(E)** Gene set enrichment analysis (GSEA) using the Hallmark gene sets highlights the top differentially regulated pathways in EBOV- (orange) and MARV-infected (blue) proximal HIOs at 3 dpi relative to mock controls. Solid bars represent significantly enriched pathways ($p < 0.05$); striped bars indicate non-significant enrichment.

mock-infected HCOs responded robustly to forskolin stimulation, demonstrating a significant increase in size. Quantitative assessment of cross-sectional area (CSA) confirmed this observation (Fig 8C). Morphologically, mock-infected organoids across both models exhibited thin epithelial walls and a defined central lumen, which expanded in response to forskolin. This response was absent in infected HCOs, despite the presence of similar baseline architecture.

These findings demonstrate that filovirus infection impairs CFTR-dependent fluid secretion in a region-specific manner. Infected HIOs retained partial responsiveness to forskolin stimulation, while infected HCOs exhibited a complete loss of swelling capacity, indicating a pronounced impairment in cAMP-dependent epithelial function in the distal colon. Under mock-infected conditions, morphological imaging revealed changes in organoid architecture: both HIOs and HCOs displayed thin epithelial walls and a central lumen, which expanded following forskolin treatment. In both, EBOV- and MARV-infected organoids, HIOs exhibited a significantly more pronounced swelling response compared to HCOs. While HIOs model the proximal intestine with architecture specialized for nutrient absorption and ion transport, HCOs represent the distal colon and are characterized by a denser population of goblet cells and a more complex mucosal barrier [19,34,35] (Fig 8D). These regional distinctions in morphology and function likely contribute to the differential impact of EBOV and MARV infection on fluid homeostasis along the intestinal tract.

### Primary cell-derived colonic epithelial organoids are permissive to EBOV and MARV infection and replication

To benchmark our findings against primary cell-derived colonic epithelial organoids (PCOs), we utilized organoids derived from two healthy adult male donors, following standard culturing, expansion, and maintenance protocols [36] (S4A Fig). PCOs were derived from two genetically distinct donors, with three or more independent experimental replicates per donor to ensure biological reproducibility. Similar to iPSC-derived HIOs and HCOs, PCOs exhibited typical multicellularity, comprising both secretory and absorptive epithelial cell types, mirroring human intestinal physiology. IHC analysis using hematoxylin (blue) and 3,3'-diaminobenzidine-based antibody staining (brown) revealed the presence of intestinal epithelial markers. Staining for chromogranin A (CHGA, enteroendocrine marker), lysozyme (LYZ, Paneth cell marker), mucin 2 (MUC2, goblet cell marker), and villin (VIL, brush border component) confirmed a fully differentiated intestinal epithelium (S4B Fig). Magnified images demonstrated CHGA localized near the basement membrane, LYZ in cytoplasmic granules, MUC2 at the apical surface, and villin distributed across the brush border, consistent with the characterization of iPSC-derived HCOs [36–38].

To evaluate the permissiveness of PCOs to EBOV infection, organoids were infected with EBOV-ZsGreen at different MOIs, and infection progression was monitored using live-cell fluorescence microscopy over 6 days. As the MOI increased, the number of EBOV-ZsGreen-positive cells rose, indicating viral replication and spread. At 2 dpi, fluorescent signals were minimal and localized to small punctate regions (S4C Fig). Robust EBOV-ZsGreen infection was observed at high MOIs, as evidenced by numerous ZsGreen-positive columnar basal cells throughout the epithelial monolayer. In contrast, mock-infected PCOs exhibited no ZsGreen-positive cells under the same fluorescence exposure (25 ms). By 6 dpi, viral spread was evident, and infected cells exhibited significant CPE, including structural changes leading to the collapse of PCO spheres. Dose- and time-dependent increases in infection efficiency were observed across MOIs ranging from 0.1 to 100, with MOI 100 reaching saturation of EBOV ZsGreen expression by 6 dpi (S4D Fig). Viral spread in PCOs was slower and less efficient compared to iPSC-derived HIOs and HCOs, with variability in infection levels across donors, suggesting donor-dependent differences in susceptibility.

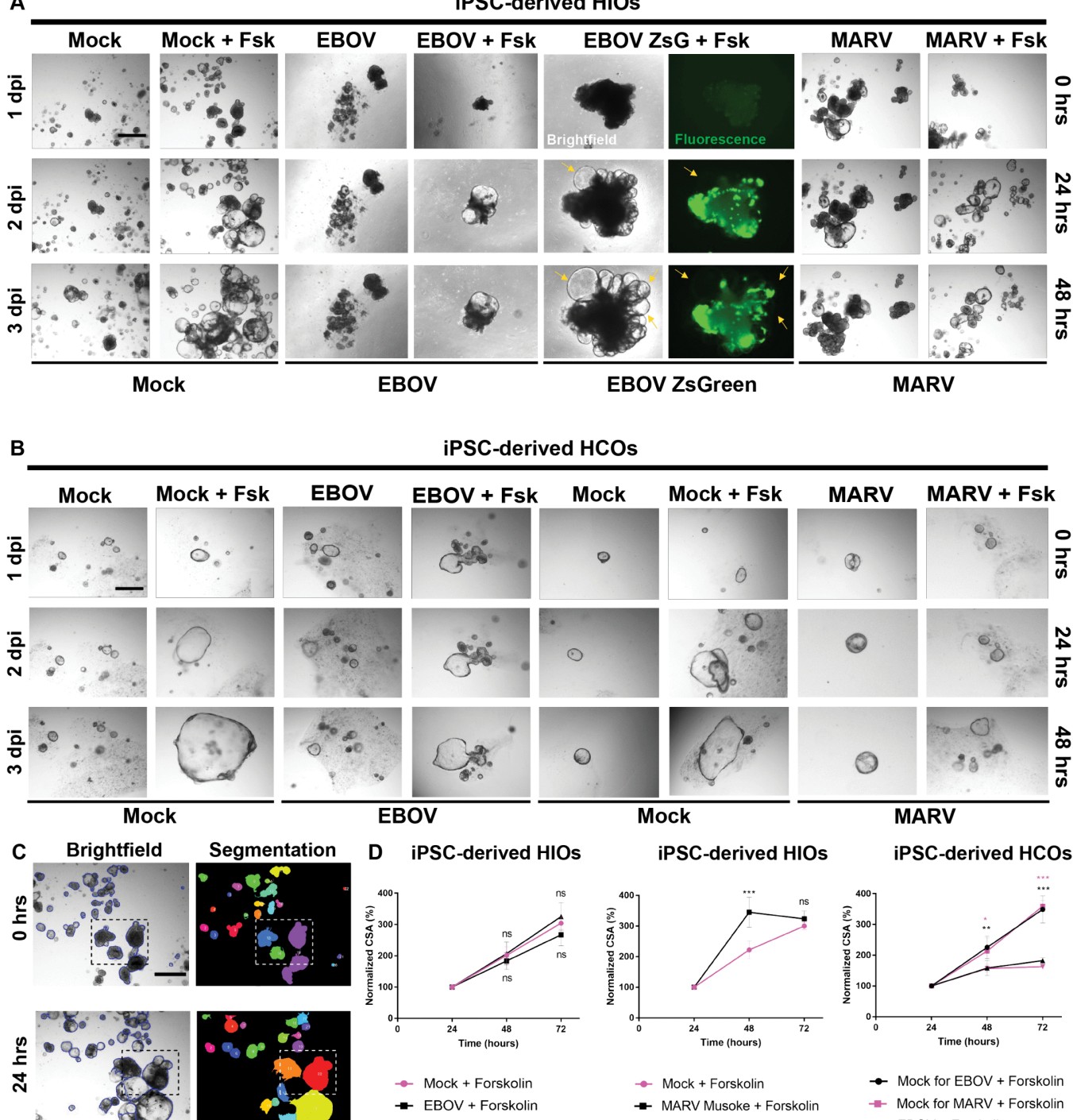

**Fig 8. Differential impact of EBOV and MARV on cAMP-mediated function in human gut organoids reveals region-specific loss of forskolin-induced swelling in HCOs. (A)** iPSC-derived HIOs were infected with EBOV, EBOV-ZsGreen, or MARV at an MOI of 10 on day 35 of differentiation. At 1 dpi, HIOs were treated with 5 μM forskolin dissolved in DMSO or treated with DMSO and imaged immediately prior to treatment (0 hours, 1 dpi), and then at 24 hours (2 dpi), and 48 hours (3 dpi) post-treatment. Time points on the left indicate days post-infection, and time points on the right indicate hours post forskolin treatment. Yellow arrows indicate organoids exhibiting visible swelling responses. **(B)** iPSC-derived HCOs were infected with EBOV or MARV at an MOI of 10 on day 35 of differentiation. At 1 dpi, organoids were treated with 5 μM forskolin dissolved in DMSO, with

DMSO-treated organoids serving as vehicle controls. Organoids were imaged at 0-, 24-, and 48-hours following treatment, corresponding to 1, 2, and 3 dpi, respectively. Time points on the left of the panel indicate days post-infection, while those on the right indicate hours post-forskolin treatment. **(C)** Automated imaging analysis of iPSC-derived HIOs, before (top) and after forskolin stimulation (bottom), was performed using OrganoSeg software to quantify changes in cross-sectional area (CSA). **(D)** Quantification of forskolin-induced swelling in HIOs and HCOs. Spheroid number and CSA were calculated using automated analytical software and normalized CSA was used to assess the mean change in organoid size following forskolin treatment. Quantification of forskolin-induced swelling is presented across three independent differentiations and three separate wells per experimental condition. Images were captured using the EVOS M5000 Imaging System, with brightfield and fluorescence imaging at 25 ms exposure. Scale bars represent 100 μm. Data are presented as mean ± standard error of the mean (SEM). Statistical analysis was performed in GraphPad Prism using two-way ANOVA with Tukey's multiple comparisons test (95% confidence interval). Asterisks indicate statistically significant differences compared to the corresponding mock-infected controls (***$p \leq 0.001$, **$p \leq 0.01$, *$p \leq 0.1$; ns, not significant).

To identify the intestinal epithelial cell types permissive to EBOV infection, PCOs were infected with EBOV at an MOI of 10 and analyzed at 3 dpi. We examined colocalization of EBOV protein VP35 (yellow) with epithelial markers including MUC2 goblet cell, VIL brush border component, CHGA enteroendocrine cell, and LYZ Paneth cell (magenta), along with DAPI (grey) for nuclei (S5A-S5D Fig). Colocalization was observed exclusively between EBOV VP35 and VIL, indicating that EBOV preferentially infects enterocytes, the predominant absorptive cell type. The colocalization occurred predominantly at the apical brush border of enterocytes (S5B Fig). Minimal colocalization with goblet cells, enteroendocrine cells, and Paneth cells likely reflects their lower abundance and the limitations of 2D sectioning in detecting colocalization in other cell types or regions.

To assess the functional capacity of PCOs for disease modeling and therapeutic applications, we performed the forskolin swelling assay. PCOs were treated with various forskolin concentrations, and swelling was monitored via microscopy. PCOs exhibited a robust, dose-dependent swelling response upon forskolin stimulation at 24 and 48 hours, demonstrating functional CFTR activity and confirming their physiological competence in ion transport regulation. (S6A Fig). PCOs were infected with EBOV or MARV at varying MOIs (0.1, 10, and 100) and treated with forskolin to assess CFTR-mediated ion transport and epithelial function. Mock-infected PCOs showed a clear response to forskolin treatment, with noticeable swelling. At an MOI of 0.1, swelling was pronounced in both EBOV- and MARV-infected PCOs, while at an MOI of 10, swelling was less evident, and no significant swelling was observed at an MOI of 100 for either virus. These results suggest that higher infection rates impair the organoids' ability to respond to forskolin, indicating that severe infection disrupts the functional integrity of the organoids and their capacity to maintain cAMP signaling-dependent responses (S6B and S6C Fig), similarly to what we observed in infected iPSC-derived HCOs. These results demonstrate that increasing viral load progressively impairs CFTR-mediated swelling responses in PCOs, reflecting a loss of epithelial ion transport function during severe filoviral infection.

## Discussion

This study establishes a regionally patterned, human iPSC-derived gut organoid platform as a robust and physiologically relevant in vitro system for investigating filoviral gastrointestinal pathogenesis. By recapitulating both proximal (small intestine-like) and distal (colon-like) epithelial lineages, these organoids enable high-resolution, region-specific interrogation of human intestinal host-pathogen interactions not possible in current filovirus disease animal models. The HIO and HCO models maintain sustained 3D architecture, support productive replication of EBOV and MARV, and exhibit hallmark features of gastrointestinal disease, including epithelial barrier disruption, dysregulated immune signaling, and transcriptional reprogramming.

Previous studies have implicated the EBOV delta peptide in mediating intestinal damage through in vivo models, most notably the murine ligated ileal loop model. In this system, delta peptide administration caused significant fluid accumulation, villous architectural disruption, and epithelial injury, indicating potent enterotoxic activity [39]. Delta peptide is a cleavage product of soluble glycoprotein (sGP), produced by orthoebolaviruses and cuevaviruses, but not by orthomarburgviruses [40], suggesting that other mechanisms also contribute to intestinal damage in addition to delta peptide, as

MARV-infected individuals evidence significant intestinal pathology. However, efforts to investigate GI damage caused by filovirus infection in human systems are limited by challenges such as post-mortem tissue autolysis and insufficient cellular resolution, particularly in the GI tract [10,41]. In contrast, our human iPSC-derived gut organoid platform enables longitudinal, spatially resolved interrogation of host-pathogen interactions in a physiologically relevant context. Notably, key clinical features of filoviral disease, such as secretory diarrhea, a major contributor to mortality during the 2014–2016 West African outbreak, are not consistently reproduced in animal models including NHPs [42–45], underscoring the need for human-specific cellular models that more accurately recapitulate gastrointestinal pathophysiology.

Our findings suggest that MARV infection of HIOs and HCOs activate cytoprotective stress responses. Previous studies show that stabilization of Nrf2 during MARV infection leads to activation of the Keap1-Nrf2 pathway, driving antioxidant and cytoprotective gene expression [46,47], which may help preserve epithelial architecture. These findings suggest that MARV employs distinct strategies to limit host cell injury.

Both iPSC-derived HIOs and HCOs, and PCOs supported EBOV and MARV infection, demonstrating broad filovirus tropism across intestinal epithelial lineages. In PCOs, enterocytes, the main nutrient absorbers and barrier maintainers, were the primary infected cells. Although replication kinetics in PCOs were slower, likely due to differences in epithelial maturity, receptor expression, or proliferation rate, viral replication remained robust.

Canonical filoviral immune features include suppression of type I interferon signaling and delayed ISG induction [48,49]. Our transcriptomic analyses revealed more pronounced ISG induction in MARV-infected organoids compared to EBOV. This difference is likely due to distinct immune evasion strategies: MARV VP35 is less effective at antagonizing RIG-I-like receptor signaling and type I interferon production than EBOV VP35, owing to structural and RNA-binding differences in their interferon inhibitory domains [50]. At 3 dpi, EBOV- and MARV-infected HCOs showed significant upregulation of immune-related gene programs, including enrichment of type I and III interferon signaling, and increased ISG expression. Immunohistochemistry and in situ hybridization at 3 dpi confirmed greater expression of *MX1*, *IFNb*, and *CXCL10* mRNA in MARV-infected tissues compared to EBOV, particularly in distal HCOs (S3 Fig). Consistent with previous studies, this interferon response likely arises from non-infected bystander cells rather than directly infected cells [30–32], emphasizing their role in shaping the immune response.

These immune evasive effects coincided with apoptotic gene activation and suppression of DNA repair pathways. We also observed epithelial barrier disruption and downregulation of oxidative phosphorylation, suggesting a conserved viral strategy to suppress host defenses and reprogram mitochondrial metabolism [51–54]. The reduced oxidative phosphorylation in infected distal HCOs and proximal HIOs may reflect a metabolic shift toward glycolysis, supporting increased biosynthetic demand during viral replication. Alterations in glucose uptake and fatty acid metabolism pathways further highlight host metabolic pathways as potential therapeutic targets.

Immunofluorescence and histochemical analyses showed that both viruses compromise tight junctions and disrupt apical architecture, implicating epithelial breakdown as a key contributor to gastrointestinal pathology. This has been supported by the transcriptomic data, suggesting disruption of key epithelial processes, such as apical polarity, tight junction integrity, and metabolic homeostasis, and is consistent with prior reports of delayed immune activation and structural epithelial damage during filoviral infection [55]. Proinflammatory and EMT pathways, which have been implicated in systemic disease progression and viral dissemination [31,56], were upregulated in both distal HCOs and proximal HIOs, suggesting that inflammatory signaling might contribute to epithelial dysfunction, as supported by EVD autopsy findings [20,57]. This highlights the intestinal epithelium as an active participant in filoviral pathogenesis.

We observed activation of hypoxia-associated transcriptional programs, including *VEGF, BNIP3L*, and *LOX,* concurrent with loss of tight junction proteins *ZO-1*, *EpCAM*, and apical membrane disorganization in both organoid types. Transcriptomic analysis revealed elevated expression of hypoxia-responsive genes in EBOV- and MARV-infected distal HIOs. This mirrors observations in other EBOV-affected tissues, such as the eye [58], and supports a conserved mechanism of barrier destabilization. Downregulation of apical polarity and tight junction markers in the organoids align with the clinical

presentation of high-volume secretory diarrhea in filovirus infection [20,59]. VEGF activation coincided with barrier desta-bilization, and chronic inflammation likely amplifies hypoxic stress, exacerbating epithelial injury [60]. Collectively, these findings propose a potential mechanistic link between viral replication, epithelial inflammation, hypoxia signaling, and barrier breakdown in filovirus gastrointestinal disease.

It is important to note that iPSC-derived intestinal organoids undergo ongoing differentiation and maturation during culture, even in the absence of infection. This temporal progression—from a proliferative, stem-like state toward mature epithelial cell types—is marked by increased expression of gut lineage-specific markers such as FABP1, MUC2, ALPI, and TFF3 [61,62]. Additionally, organoid development is influenced by culture conditions, including growth factors and extracellular matrix components, which support continued epithelial maturation. These changes are consistent with previ-ously reported increases in enterocyte and goblet cell markers during early organoid development [61] and sex- or strain-specific differences in gene expression during differentiation [63]. To account for these dynamics, all infection-induced gene expression changes in our study were analyzed relative to time-matched mock-infected controls, ensuring that the effects attributed to filoviral infection are distinct from baseline developmental processes.

Functional assessment of ion transport revealed region-specific disruptions following filoviral infection. Proximal HIOs maintained cAMP-dependent signaling, as shown by normal forskolin-induced swelling, whereas distal HCOs failed to respond, indicating a profound impairment of the adenylate cyclase–PKA–CFTR axis [33]. PCOs, derived from human colonic tissue, also exhibited a lack of swelling in response to infection, consistent with the functional impairment observed in iPSC-derived HCOs and underscoring the heightened susceptibility of the distal colon to filoviral infection. Given that HIOs and HCOs model distinct segments of the human GI tract, these findings underscore how viral modulation of epithe-lial function varies across anatomical sites, with potential implications for disease severity and dissemination.

Transcriptomic and physiological data also revealed ion transporter dysregulation, including Na$^+$/K$^+$ATPase and sodium–proton exchangers, along with increased epithelial stress and apoptosis. These findings mechanistically align with the clinical presentation of high-volume secretory diarrhea in filoviral disease [20,59,64,65] and reinforce the model's utility in replicating key features of human gastrointestinal dysfunction. The selective susceptibility of distal organoids high-lights a central role for cAMP in both fluid regulation and epithelial proliferation and survival [66]. The preserved swelling response in HIOs, despite severe disruption in HCOs, aligns with clinical EVD [20,67], and support the idea that dysregu-lated fluid transport drives diarrhea, identifying potential therapeutic targets for restoring barrier function and signaling [68].

In conclusion, we have established a robust, regionally patterned, human iPSC-derived gut organoid model as a power-ful platform for dissecting the gastrointestinal pathogenesis of EBOV and MARV. By overcoming limitations of traditional in vivo and ex vivo filovirus intestinal disease models, including post-mortem tissue autolysis and limited cellular resolu-tion, this system enables reproducible, high-resolution analysis of epithelial-intrinsic responses under BSL-4-compatible conditions. The model captures key features of filoviral gastrointestinal disease, including viral replication, immune dysreg-ulation, metabolic reprogramming, and barrier breakdown, and reveals region-specific dysfunction, especially in the distal colon, linked to high-volume diarrhea. Furthermore, the system offers a scalable platform for therapeutic screening and targeted intervention strategies aimed at restoring epithelial homeostasis.

## Limitations of the study

Despite its advantages, the model has several limitations. It lacks immune, stromal, vascular, and microbial components, which are critical for capturing the full complexity of host-pathogen interactions. Additionally, the organoids represent a fetal-like state of epithelial maturation, which may not fully recapitulate adult intestinal physiology. Our model is optimized for acute infection and does not capture long-term sequelae such as viral persistence, chronic inflammation, or post-infectious remodeling—hallmarks of filoviral disease progression [57].

We also focused on single isolates of EBOV (Mayinga) and MARV (Musoke); while well-characterized, responses may vary with other viral variants, warranting further investigation. A practical limitation is the absence of infectious viral

titer measurements in supernatants. Due to the 3D structure of organoids and biosafety constraints in BSL-4 settings, viral release into the media is often restricted. Future work incorporating optimized titration and sampling methods will be needed to directly assess extracellular viral load.

Another inherent limitation involves the number of donor lines. While our study used two genetically distinct iPSC lines and validated findings in primary tissue-derived organoids from independent donors, expanding donor diversity would improve generalizability by accounting for inter-individual variation. However, the complexity of long-term differentiation protocols and BSL-4 restrictions limited our ability to include more lines. Importantly, our original design included additional iPSC donor lines (including a female donor), but technical limitations precluded successful differentiation into mature 3D gut organoids for some lines—an outcome consistent with well-documented variability in iPSC differentiation potential [69]. To mitigate this, we performed multiple independent differentiations per line and applied orthogonal validation techniques (e.g., immunofluorescence, IHC, ISH, RNA-seq) to ensure biological replication and highlight robust phenotypes unlikely to be donor-specific artifacts [70]. Nonetheless, future studies incorporating more diverse donor lines will be critical for further validation.

Despite these limitations, our iPSC-derived gut organoid platform represents a significant advance for modeling human-specific filovirus pathogenesis. Its scalability, physiological relevance, and adaptability lay a strong foundation for future co-culture models incorporating additional cellular components to better replicate in vivo complexity. As such, it holds substantial promise for elucidating antiviral defense mechanisms, guiding therapeutic development, and improving our preparedness against emerging high-consequence viral pathogens.

## Materials and methods

### Ethics statement

All differentiation experiments were performed using de-identified iPSC lines with approval by Boston University's Institutional Review Board.

### Biosafety considerations

All work with wildtype and recombinant EBOV and MARV was performed in the BSL-4 facility of Boston University's National Emerging Infectious Diseases Laboratories (NEIDL) following standard operating procedures approved by the Boston University Institutional Biosafety Committee.

### Differentiation and maintenance of human intestinal organoids (HIOs) and human colonic organoids (HCOs)

HIOs and HCOs were derived from human induced pluripotent stem cells (iPSCs). Only previously reprogrammed human iPSC lines were used for this work. Parental iPSC lines, including the BU1 CG iPSC line with a CDX2/GFP reporter (SPC2-ST-B2 clone), were previously published by our group. The BU1 CG line, derived from a healthy male donor, has a normal karyotype (46XY). iPSCs were maintained in feeder-free conditions on growth factor-reduced Matrigel (Corning cat. no. 354277), using mTESR1 (StemCell Technologies), and passaged on hESC Matrigel (Corning) with ReLeSR (Corning) reagent as per manufacturer instructions. Pluripotent stem cell lines used in this study, along with maintenance standard operating procedures and directed differentiation protocols, are available from the CReM iPSC Repository at Boston University and Boston Medical Center and can be found at (https://crem.bu.edu/cores-protocols/).

At >90% confluency, iPSCs were differentiated into HIOs using an established protocol [17]. iPSC colonies were dissociated with Gentle Cell Dissociation Reagent (StemCell Technologies), re-plated at $2 \times 10^6$ cells per well on Matrigel-coated six-well plates in mTeSR1 with Y27632 (5 µM) and differentiated into definitive endoderm with the StemDiff Definitive Endoderm Kit. Cells were analyzed by flow cytometry using anti-CXCR4-PE and anti-c-kit-APC antibodies. On day 3, cells were split 1:3 into new hESC Matrigel-coated plates and treated with DS/SB43 (Dorsomorphin and

SB431542) and Y27632, then with DS/SB without Y27632. On day 6, cells were split again and cultured in CB/RA medium with CHIR99021, rhBMP4, and retinoic acid. Both DS/SB and CB/RA media were based on complete serum-free differentiation medium (cSFDM). A comprehensive list of reagents and catalog numbers, media recipes, and antibodies can be found in the previously published [17]. For all infection experiments, HIOs were derived from two genetically distinct iPSC lines, each subjected to three or more independent differentiations per line to ensure biological replication and assess reproducibility. A detailed protocol of HCO and HIO differentiation was deposited in protocols.io [71].

### Differentiation of primary cell-derived colonic organoids (PCOs)

PCOs were established from adult human colonic tissue. The primary cell-derived colonic organoids used in this study were kindly provided by Dr. Ryan B. Corcoran, Massachusetts General Hospital, Boston MA. Two lines derived from de-identified healthy adult male donors were utilized.

Organoid Thawing and Plating: Organoids were thawed by removing the cryovial containing the organoids from the freezer and immediately placing it into a 37°C water bath. The cryovial gently swirled for 1–2 minutes until fully thawed. Once thawed, the cryovial was carefully removed from the water bath. Using a sterile pipette, the contents of the cryovial were transferred to a sterile 15 mL conical tube containing pre-warmed selection medium. The suspension was then centrifuged to remove the cryoprotectant. After the aspiration of the supernatant, the organoids were resuspended in fresh selection medium and plated for further culture and maintenance.

Basal Growth Medium (BGM) Preparation: BGM was prepared by combining WNT medium, R-spondin medium, DF20, and necessary supplements as outlined in the Basal Media Growth factors (S1 Table), with DF20 added to increase FBS content.

Organoid Feeding: Medium was replaced every 3–4 days or more frequently if it turned yellow. Media changes were most frequent before passaging and least frequent afterward. Old media was aspirated, and fresh media was added without disturbing the Matrigel dome. Volumes used were 1 mL for 12-well plates, and 500 µL for 24-well plates.

Organoid Passaging: Organoid passaging was performed to dissociate organoids into small cell clusters and reseed them in fresh 3D Matrigel domes, renewing the stem cell population and maintaining pluripotency. Plates were placed on ice, and the BGM was aspirated. Cell Recovery Solution (CRS, Corning; 1 mL for 12-well plates) was added to each well and incubated for 15 minutes. The 3D Matrigel dome was detached by pipetting the solution around its edges and gently breaking it into smaller pieces. The mixture was transferred to a 15 mL conical tube and incubated on ice for 1 hour to aid Matrigel dissociation. After centrifugation at 200×g for 5 minutes, the supernatant was aspirated, and the cell pellet was resuspended in 3 mL TrypLE and incubated at 37°C for 5 minutes. The suspension was then passed through a 20G needle several times to further dissociate organoid clusters. To neutralize TrypLE, 10 mL of Splitting Media was added, and the sample was centrifuged again. The supernatant was aspirated, and the pellet was resuspended in 3D Matrigel (50 µL for 12-well plates). The plates were incubated for 30 minutes at 37°C to allow for Matrigel solidification and organoid formation, after which fresh cell culture medium was added to the wells. For all infection experiments, PCOs were derived from two independent adult donors. Organoids from each donor were expanded and used in three or more independent experimental replicates to ensure biological replication and assess reproducibility. A detailed protocol of PCO differentiation was deposited in protocols.io [71].

### Cell lines

African green monkey kidney cells (Vero E6; ATCC CRL-1586) were used for virus propagation. The cells were maintained in Dulbecco's modified Eagle medium (DMEM) supplemented with 200 mM L-glutamine and 10% fetal bovine serum (FBS). Cell culture medium was supplemented with 100 µg/mL Primocin. Cells were maintained at 37°C and 5% $CO_2$.

## Virus propagation

EBOV isolate Mayinga (GenBank accession number NC_002549) and MARV isolates Musoke (GenBank accession number NC_001608) were kindly provided by H. Feldmann, NIH NIAID Rocky Mountain Laboratories, Hamilton, MT. Recombinant EBOV expressing ZsGreen as a reporter protein (rEBOV-ZsGreen-2PA-VP40; based on Mayinga isolate) was generated in the Mühlberger lab [21]. Filoviruses were propagated in Vero E6 cells as described before [72]. Cell supernatants were centrifuged at 5,250 ×g for 10 min at 4°C to remove cellular debris. Clarified supernatants were purified over a 20% sucrose cushion by centrifugation at 80,000 x g for 2 hours at 4°C and viral pellets were resuspended in PBS. Viral titers were determined by tissue culture infectious dose 50 ($TCID_{50}$) assays using the Spearman Karber algorithm [73].

## Infection of organoids

**Preparation of organoids for infection.** Organoids were maintained in 12-well plates embedded in Matrigel. Prior to infection, media was carefully aspirated from each well while preserving the integrity of the Matrigel droplet. A total of 500 µL of chilled complete rinse solution (CRS) was added to each well, and plates were incubated at 4°C for 30 minutes. Following incubation, Matrigel droplets were transferred to 1.5 mL tubes using a P1000 pipette tip with the end cut off to minimize mechanical disruption. Wells were rinsed with 500 µL PBS to collect any residual organoids, which were combined with the initial suspension. The suspension was centrifuged at 200 ×g for 5 minutes, and the supernatant was aspirated, leaving a minimal volume of medium to prevent pellet loss. The organoids were then resuspended in 1 mL of the appropriate medium: CKDCI for distal HIOs, CKDI for distal-cAMP HIOs, or IMCK for proximal HIOs.

All EBOV and MARV stocks were prepared from cell culture supernatants and purified via ultracentrifugation through a 20% sucrose cushion. Viral titers were determined using the tissue culture infectious dose 50 ($TCID_{50}$) assay. Mock-infected controls were included in all experiments and processed in parallel with infected samples. These organoids underwent the same handling procedures, including dissociation, incubation, Matrigel re-embedding, media exchange, and inactivation, but without the addition of virus. The same organoid culture medium used to dilute viral stocks was applied to mock controls.

**Multiplicity of infection (MOI) calculation.** To determine the MOI, organoids from three representative wells were dissociated into single-cell suspensions using enzymatic digestion. The average cell number per well was calculated to account for variability in organoid size and density. The number of infectious units needed for each condition was calculated using the formula:

**Average cell numbers per well × MOI × number of wells to be infected.** Based on the known $TCID_{50}$ titer, the corresponding volume of virus stock was added to the appropriate organoid medium (CKDCI, CKDI, or IMCK) to achieve the desired MOI. Unless otherwise noted, infections were performed at an MOI of 10. This higher MOI was necessary to overcome diffusion barriers associated with the 3D structure of organoids and to ensure robust, reproducible infection. Additional infections were performed at MOIs of 0.1, 10, 50, and 100 to assess dose-dependent responses.

**Viral infection.** Following MOI calculation, the appropriate volume of virus was added to the organoid suspension and incubated at 37°C for 1 hour. Organoids were then washed once with 1 mL PBS and resuspended in fresh 3D Matrigel. The mixture was replated into 12-well plates and incubated at 37°C for 30 minutes to allow Matrigel polymerization. After polymerization, 1 mL of the appropriate organoid medium (CKDCI, CKDI, or IMCK) was added to each well, and plates were returned to the incubator for downstream analysis. A detailed protocol describing standardized infection of human gut organoids with EBOV and MARV was deposited in protocols.io [74].

## Live Imaging of EBOV-ZsGreen-Infected Organoids

Live-cell imaging of organoids infected with EBOV-ZsGreen was performed in a BSL-4 setting using the EVOS M5000 Imaging System (Thermo Fisher Scientific). Organoids were embedded in Matrigel and maintained in CKDCI or IMCK medium throughout the course of infection. Exposure time, illumination intensity, and all acquisition settings were

standardized and held constant across all conditions and time points to ensure data comparability. Mock-infected organoids, maintained under identical culture and imaging conditions, were included in all experiments as negative controls to account for background fluorescence and baseline morphological variation. ZsGreen fluorescence served as a direct reporter of EBOV infection and was evaluated qualitatively and, where applicable, quantitatively using Fiji. All imaging experiments were independently repeated using multiple biological replicates.

## Immunofluorescence analysis

Organoids were harvested from 3D Matrigel cultures using CRS (1 ml/well) for 30 minutes to 1 hour at 4°C. After digestion, organoids were washed with PBS to remove residual CRS. For larger organoids, settling was allowed for 10 minutes without centrifugation, while smaller organoids were briefly centrifuged for 2 minutes at <200 x g and 4°C. Non-infected BSL-2 samples were fixed in 4% paraformaldehyde (PFA) for 30 minutes at room temperature, followed by washing with PBS. Infected and non-infected BSL-4 samples were fixed for at least 6 hours with PFA following approved inactivation SOPs before removal from the BSL-4 lab. After fixation, organoids were transferred to individual Eppendorf tubes and briefly centrifuged for 2 minutes at <200 x g at room temperature after each step to pellet, ensuring minimal damage. For permeabilization, organoids were incubated in PBS-Tx (PBS + 0.5% Triton X-100) for 15 minutes at room temperature. Blocking was performed by incubating the organoids for 60 minutes at room temperature in 500 µL of blocking buffer (PBS-Tx + 4% goat serum) while gently rocking. Primary antibody incubation was performed overnight at 4°C using the respective primary antibodies (S2 Table) diluted in blocking buffer. After the primary antibody incubation, organoids were washed at least three times in PBS-Tx for 30 minutes per wash on a rotating shaker. Secondary antibodies (S2 Table) were diluted in PBS and incubated for one hour at room temperature. Following the incubation, organoids were washed at least three times in PBS-Tx, with each wash lasting 30 minutes on a rotating shaker.

Nuclear staining was carried out by incubating organoids with Hoechst 33342 (1:2,000 dilution) for 10 minutes at room temperature. Hoechst was preferred over DAPI to avoid edge effects on organoid imaging. After the final wash in PBS, the organoids were resuspended in up to 50 µL of ProLong Diamond Antifade Mountant (without DAPI) and applied to the center of a cavity slide. A coverslip was placed on top, ensuring the slide was flipped onto the coverslip, allowing the organoids to sink toward it and preventing them from settling at the bottom of the cavity, which would impair imaging. Organoids were imaged using a Zeiss LSM 710-Live Duo Confocal microscope with two-photon capability at 10x, 20x and 63x for high resolution visualization.

## EBOV infection rate of HCOs determined by FACS analysis

The NIR Live/Dead staining and FACS analysis were performed as follows: NIR dye was reconstituted in DMSO, wrapped in aluminum foil to protect from light, and stored at -20°C. A working solution of NIR Live/Dead stain was prepared by diluting the stock in PBS. FACS buffer was prepared by adding 0.5% PBS and 2 mM EDTA. Organoids were dissociated by adding 1 mL of CRS to each well containing the 3D Matrigel pellet, followed by a 30-minute incubation at 4°C. The Matrigel droplets were then transferred to 1.5 mL tubes using a P1000 pipette with the tip cut off to minimize dissociation. The wells were washed with 500 µL PBS to collect remaining cells, and the suspension was centrifuged at 200 × g for 5 minutes. The supernatant was aspirated, and the pellet was resuspended in 1 mL of the appropriate medium (CKDCI for distal HIOs, CKDI for distal-cAMP HIOs, or IMCK for proximal HIOs). The organoid suspension was then stained with 1:800 diluted NIR Live/Dead stain in 250 µL PBS, mixed thoroughly, and incubated for 30 minutes at room temperature, protected from light. After incubation, the cells were washed with FACS buffer and resuspended in 4% paraformaldehyde (PFA) for a minimum of 6 hours, following the formalin/aldehyde BSL4 inactivation protocol. After inactivation, the samples were transferred to a BSL2 setting for further processing. In the BSL2, the cells were washed with FACS buffer, permeabilized with 0.5% Triton X-100 in PBS for 15 minutes, and blocked with 250 µL of 5% goat serum diluted 1:100 in PBS for

15 minutes. After blocking, the cells were washed and incubated with the primary antibody, diluted in blocking buffer, for 1 hour at room temperature in the dark. Following primary antibody incubation, the cells were washed with FACS buffer and incubated with the secondary antibody diluted in PBS for 1 hour at room temperature. The used antibodies are listed in S2 Table. The cells were washed three times with FACS buffer and resuspended in 350 μL FACS buffer. The samples were analyzed by FACS on a Stratedigm machine, with gating performed first on live/dead cells and then on infection status. Appropriate mock controls were included for all experimental conditions, including mock unstained, virus unstained, mock + NIR, virus + NIR, mock + virus secondary antibody only, virus + virus secondary antibody only, and mock or infected cells with virus antibody at a 1:200 dilution.

## Histology and immunohistochemistry

Organoids were fixed in 10% neutral-buffered formalin for 72 hours in compliance with institutional standard operating procedures. Organoids were placed in HistoGel (Epredia, Kalamazoo, Michigan, USA) and processed routinely as formalin fixed paraffin embedded (FFPE) blocks. 5 μm sections were cut for subsequent staining. IHC was performed on a Ventana Discovery Ultra autostainer (Roche Diagnostics, Indianapolis, IN, USA). Pretreatment was performed with Benchmark Ultra CC1, a Tris-based antigen retrieval buffer, at 95°C for 32 minutes. Pre-diluted secondary HRP polymer antibodies were used for developing all primary antibodies (MP-7451 or MP-7452, Vector Laboratories, Newark, CA, USA) for 20 min at 37°C following a protein blocking step with Akoya Opal Diluent/Block (ARD1001EA, Akoya Biosciences, Marlborough, MA, USA). Slides were counterstained with hematoxylin or DAPI and coverslipped with Micromount mounting media (Leica, Wetzlar, Germany) or Prolong Gold Antifade Mountant (Invitrogen, Waltham, MA, USA). Information about the used antibodies and parameters for ISH and IHC assays are provided in S3 Table. Whole slide-images were acquired with a PhenoImager HT Automated Quantitative Pathology Imaging System, which included onboard unmixing to minimize background signal (Akoya Biosciences). Exposures for all Opal dyes on the Vectra were set based upon regions of interest with strong signal intensities to minimize exposure times and maximize the specificity of signal detected.

## Quantitative image analysis

Fluorescent ISH-IHC images were analyzed using HALO software (v4.05107.357; Indica Labs, Inc., Corrales, NM). View settings were optimized to enhance the visibility of immunomarkers and reduce background noise by adjusting threshold gates for minimum signal intensity. Organoid regions were annotated using the MiniNet classifier in HALO AI (Indica Labs). The classifier was trained on several organoids via manual annotations, and after it was run across all fluorescently labeled slides, its automated annotations were reviewed in detail. Any inaccurate annotations were manually corrected or removed. To quantify signal from *MX1*, *IFNb*, *CXCL10* probes, or viral immunoreactivity, we used the HALO Area Quantification (AQ) module (v2.4.9). The algorithm was customized to detect positive fluorescent signal based on specific color and intensity parameters for each target, and it was run across annotations made by the MiniNet classifier. The module output the percentage of the annotated slide area that displayed positive signal. The resulting data were exported as a.CSV file and analyzed in GraphPad Prism (v10.2.0; Dotmatics, San Diego, CA).

For brightfield IHC with DAB chromogen, whole-slide images were likewise analyzed in HALO using the brightfield Area Quantification (AQ) module (v2.4.9). The algorithm was configured to separate DAB (positive stain) from hematoxylin (counterstain) using color deconvolution, and a positivity threshold was established based on representative control tissues. Organoid regions were annotated with the flood-selection tool, with any inaccurate annotations were manually corrected or removed, and only annotated areas were included in analysis. Within each annotation, the module calculated the proportion of tissue area positive for DAB signal. These values were exported as summary statistics (percent positive tissue area) and used for downstream comparisons in GraphPad Prism. For all analyses, including those shown in S3 Fig, multiple images were quantified per biological replicate. Specifically, at least five representative fields of view per sample

(n = 3 biological replicates) from two independent infections were analyzed to ensure robust and reproducible sampling and to avoid potential bias from relying on single representative images.

## RNA isolation and library preparation

Organoids infected with EBOV or MARV were harvested for RNA analysis at 1 and 3days dpi. To harvest organoids, culture medium was carefully aspirated from each well while maintaining the integrity of the 3D Matrigel domes. A volume of 500 µL chilled CRS was added to each well, and plates were incubated at 4 °C for 30 minutes to facilitate Matrigel dissociation. The resulting suspension was transferred to 1.5 mL tubes. Wells were subsequently rinsed with 500 µL of PBS to recover any residual organoids or cellular debris, which was combined with the corresponding sample.

Samples were centrifuged at 200 × g for 5 minutes at 4 °C, and the supernatant was carefully aspirated. 1 mL of TRIzol reagent (Thermo Fisher Scientific) was added directly to the pellet, followed by a 10-minute incubation at room temperature. Samples were vortexed thoroughly to ensure complete homogenization. Inactivated samples were transferred to the BSL-2 laboratory in accordance with approved institutional inactivation SOPs.

Total RNA was isolated from human iPSC-derived organoids using TRIzol reagent (Thermo Fisher Scientific), following the manufacturer's protocol. Briefly, 200 µL of chloroform was added per 1 mL of TRIzol, followed by centrifugation at 12,000 × g for 15 minutes at 4 °C to separate the phases. The aqueous phase was collected, and RNA was precipitated with isopropanol, washed with 75% ethanol, and resuspended in nuclease-free water. RNA concentration. Residual genomic DNA was removed by DNase I treatment. RNA-seq libraries were prepared using a poly(A) selection strategy to enrich for mRNA. Libraries were sequenced on an Illumina NextSeq 2000 platform using a P3 flow cell (2 × 50 bp paired-end reads), generating approximately 1.1 billion total reads per run.

## RNA sequencing data analysis

The quality of the raw data was assessed using FastQC v.0.11.7 [75]. The sequence reads were aligned to the GRCh38 reference with added sequences of EBOV and MARV using STAR v.2.6.0 [76]. Counts per gene were summarized using the featureCounts function from the subread package v.2.0.3 [77]. The edgeR package v.4.2.0 [78] was used to import, organize, filter and normalize the counts and the matrix of counts per gene per sample was then analyzed using the limma/voom normalization method [79]. Genes were filtered based on the standard edgeR filtration method using the default parameters for the "filterByExpr" function, which excludes genes with low expression across all samples. Specifically, genes with fewer than 10 counts were removed from the dataset prior to differential expression analysis. After exploratory data analysis with Principal Component Analysis (PCA), contrasts for differential expression testing were done for each of the infected samples vs mock-infected controls at each time point. The limma package v.3.60.0 [79] with its voom method, namely, linear modelling and empirical Bayes moderation was used to test differential expression (moderate t-test). P-values were adjusted for multiple testing using Benjamini-Hochberg correction (false discovery rate-adjusted p-value; FDR). Differentially expressed genes for each comparison were visualized using Glimma v.2.14.0 [80], and FDR < 0.05 was set as the threshold for determining significant differential gene expression. Functional predictions were performed using the fgsea v.1.30.0 package [81] for gene set analysis.

The RNA-seq data have been deposited in the Gene Expression Omnibus (GEO) under accession numbers **GSE298600** and **GSE300073**.

## Forskolin-induced stimulation assay

HIOs were dissociated as described in the infection section. Following dissociation, the HIOs were washed with PBS and centrifuged at 200 × g for 3 minutes. The resulting cell pellet was resuspended in fresh 3D Matrigel at 37°C, ensuring a low density to prevent significant spheroid overlap and facilitate downstream imaging and analysis. The resuspended

HIOs were plated as droplets onto pre-warmed 37°C tissue culture plates, coated with medium, and allowed to recover for 48–72 hours. The CKDCI medium was supplemented with either forskolin or DMSO control. After 24 hours, high-resolution images of the spheroids were captured using the EVOS M50000 Imaging System. The organoids were then replenished with fresh CKDCI or IMCK medium containing either the 5 µM forskolin (Sigma, catalog #F3917) or vehicle control. After 24 hours, images were acquired and analyzed using the same parameters applied to the pre-forskolin (baseline) images. This procedure was repeated at each designated time point 24, 48, 72 hours post treatment. All images were saved in .tif format and analyzed using OrganoSeg, an open-source MATLAB plug-in previously described [82]. To quantify the cross-sectional area (CSA) of all organoids per well (referred to as "whole-well CSA"), images were segmented using the following parameters: intensity threshold of 0.5, window size of 100, and size threshold of 100. Post-segmentation images were manually reviewed for quality control, excluding spheroids that were on the edge of the field, had burst or flattened, or were not present in both pre- and post-treatment images. OrganoSeg provided output metrics including total sphere number and CSA. "Normalized" whole-well CSA was calculated by comparing pre- and post-forskolin images using the equation: [Whole well CSA post/Whole well CSA pre x100]. Control wells, treated with DMSO (vehicle for forskolin) and/or intestinal media, were included in each experiment. The CSA measurements from control wells were subtracted from those of experimental wells. Each independent experiment included at least three replicate wells per treatment condition. Statistical analyses were performed using GraphPad Prism. Comparisons between treatment groups were conducted using two-way ANOVA followed by Tukey's multiple comparisons test with a 95% confidence interval. Data are presented as mean ± standard error of the mean (SEM), and p-values ≤ 0.05 were considered statistically significant.

## Supporting information

**S1 Table. Composition of culture medium for primary colonic organoids (PCOs).** Detailed list of all components used in the culture medium for establishing and maintaining primary colonic organoids. The table includes reagent name, final concentration, supplier, catalog number, and notes on growth factor or supplement function where applicable.
(DOCX)

**S2 Table. Antibodies, dyes, and reagents used for immunofluorescence, immunohistochemistry, and flow cytometry.** Comprehensive list of all antibodies, dyes, and reagents used in this study. Information includes target antigen, host species, clone or catalog number, source or supplier, application (e.g., IF, IHC, FACS), and dilution.
(DOCX)

**S3 Table. Antibodies used for immunohistochemistry (IHC) analysis.** List of primary and secondary antibodies used for IHC staining of primary and induced pluripotent stem cell–derived gut organoids. The table includes target antigen, antibody host species, clone or catalog number, supplier, dilution, and application details.
(DOCX)

**S1 Fig. BU310-Cre2 iPSC-derived HCOs are permissive to EBOV and MARV infection.** (A) BU310-Cre2 iPSCs were differentiated into definitive endoderm by day 3, followed by specification into CD26⁺ gut progenitors by day 15. (B) Gut progenitors were isolated by fluorescence-activated cell sorting (FACS) and cultured in CKDCI medium to generate HCOs. On day 35 of differentiation, HCOs were infected with EBOV or MARV at a MOI of 10. (C–H) Confocal microscopy was performed at 1 and 3 dpi to assess viral replication. Immunofluorescence staining was conducted using antibodies against viral nucleoproteins (NP for EBOV and NC for MARV; red), villin1 to mark intestinal epithelial cells (green), and Hoechst for nuclear counterstaining (blue). Images were acquired using a Zeiss LSM 710 Live-Duo confocal microscope with two-photon capability. Panels C, D, F, and G: scale bars = 100 µm; panels E and H: scale bars = 10 µm. Data shown are representative of three independent infection experiments (n = 3).
(TIF)

**S2 Fig. Viral transcript production in EBOV- and MARV-infected HCOs at 1 and 3days post-infection.** Loess-smoothed plots showing the relative abundance of viral transcripts in human colonic organoids (HCOs) infected with EBOV or MARV compared to mock-infected controls at 1 and 3dpi. **(A)** EBOV-infected versus mock-infected HCOs. **(B)** MARV-infected versus mock-infected HCOs. Data are representative of $n = 3$ independent infections per condition. (TIF)

**S3 Fig. Virus- and region-specific epithelial immune activation in gut organoids at 3dpi.** (A–C) Representative combined IHC and ISH images of distal HCOs and proximal HIOs infected with MARV or EBOV at an MOI of 10 and harvested at 3 dpi. Staining was performed using ISH probes targeting the mRNA of human interferon-stimulated genes (ISGs) *MX1* (A), *IFNb* (B), and *CXCL10* (C) (yellow), and antibodies targeting EBOV VP35 or GP MARV, respectively (red), with DAPI for nuclear counterstaining (gray). (D–F) Quantification of immune-reactive positive pixel area for *MX1* (D), *IFNb* (E), and *CXCL10* (F) mRNA expression. Data are representative of two independent infections (n = 3). Scale bars = 200 µm. (TIF)

**S4 Fig. Experimental overview and infection dynamics of primary cell-derived intestinal colonic (PCOs) organoids.** (A) Schematic illustrating the experimental protocol for primary cell-derived human intestinal organoids (PCOs). Created in BioRender. Muhlberger, E. (2025) https://BioRender.com/ev6enkh. (B) Immunohistochemical analysis confirms the presence of lysozyme-positive Paneth cells, chromogranin A-positive enteroendocrine cells, mucin 2-positive goblet cells, and expression of the apical brush border protein villin in PCOs. Hematoxylin staining (blue) and antibody-based staining (brown) are shown. Scale bar = 100 µm. (C and D) Organoids were infected with EBOV-ZsGreen at varying MOIs, 0.1, 10, 50 and 100 and imaged at 2 and 6 dpi. Images were captured using the EVOS M50000 Imaging System. Exposure time for fluorescence images was 25 ms. Scale bar = 100 µm. (TIF)

**S5 Fig. Selective colocalization of EBOV VP35 with enterocytes in PCOs.** PCOs were infected with EBOV at an MOI of 10. At 3 dpi, cells were fixed with 4% PFA and processed for staining using antibodies against EBOV VP35 and cell-type-specific intestinal epithelial markers. Colocalization of EBOV VP35 (yellow) with intestinal epithelial cell markers in 2D sections of primary cell-derived intestinal organoids. Intestinal epithelial cell types were identified using specific markers: (A) MUC2 (magenta, goblet cells), (B) VIL (magenta, brush border protein), (C) CHGA (magenta, enteroendocrine cells), and (D) LYZ (magenta, Paneth cells), with nuclei stained using DAPI (grey). Inset in (B) shows a magnified view of the apical brush border to highlight colocalization. Scale bar = 100 µm. (TIF)

**S6 Fig. Disruption of organoid integrity and impaired forskolin response in PCOs infected with EBOV or MARV at high MOI.** (A) At day 35 of differentiation, organoids were treated with forskolin at concentrations of 5, 10, and 20 µM (dissolved in DMSO) and imaged at 0 hours (top), 24 hours (middle), and 48 hours (bottom) to assess forskolin-induced swelling. Images were acquired using a Keyence BZ-X710 fluorescence microscope. Scale bar = 500 µm. (B and C) PCOs were infected with (B) EBOV or (C) MARV at MOI 0.1, 10, or 100. At 1 dpi, organoids were treated with 5 µM forskolin and subsequently imaged at 0 hours (1 dpi), 24 hours (2 dpi), and 48 hours (3 dpi) post-treatment. Changes in organoid morphology, including swelling and structural integrity, were evaluated at each time point. Images were captured using the EVOS M5000 Imaging System. Scale bar = 500 µm. (TIF)

## Acknowledgments

We gratefully thank Anna Pyle, Yale University for her pivotal role in securing the funding that supported this work. We are thankful to the CReM iPSC Core, Flow Cytometry Core, Cell Imaging Core, and Single Cell Sequencing Core at the Boston

University Chobanian & Avedisian School of Medicine for technical assistance. We are also grateful to Dr. Ryan Corcoran at Massachusetts General Hospital for generously providing the primary epithelial colonic organoid lines used in this study.

## Declaration of AI-assisted technologies in the writing process

During the preparation of this work, Grammarly software was used to assist with language refinement of the first draft of the manuscript. After using this tool, the authors reviewed and edited the content as needed and take full responsibility for the content of the publication.

## Author contributions

**Conceptualization:** Gustavo Mostoslavsky, Elke Mühlberger.

**Data curation:** Elizabeth Y. Flores, Judith Olejnik, Pushpinder Bawa, Feiya Wang, Aoife K. O'Connell, Anna Tseng, Nicholas A. Crossland.

**Formal analysis:** Elizabeth Y. Flores, Pushpinder Bawa, Feiya Wang, Aoife K. O'Connell, Anna Tseng, Nicholas A. Crossland.

**Funding acquisition:** Gustavo Mostoslavsky, Elke Mühlberger.

**Investigation:** Elizabeth Y. Flores, Adam J. Hume, Judith Olejnik, Aoife K. O'Connell, Anna Tseng, Nicholas A. Crossland.

**Methodology:** Elizabeth Y. Flores, Adam J. Hume, Judith Olejnik, Aditya Mithal, Andrew D'Amico, MengWei Yang, Aoife K. O'Connell, Anna Tseng, Nicholas A. Crossland.

**Project administration:** Gustavo Mostoslavsky, Elke Mühlberger.

**Resources:** Gustavo Mostoslavsky, Elke Mühlberger.

**Supervision:** Adam J. Hume, Judith Olejnik, Nicholas A. Crossland, Gustavo Mostoslavsky, Elke Mühlberger.

**Validation:** Elizabeth Y. Flores, Pushpinder Bawa, Feiya Wang.

**Visualization:** Elizabeth Y. Flores, Aoife K. O'Connell, Anna Tseng, Nicholas A. Crossland, Gustavo Mostoslavsky, Elke Mühlberger.

**Writing – original draft:** Elizabeth Y. Flores.

**Writing – review & editing:** Elizabeth Y. Flores, Adam J. Hume, Judith Olejnik, Aoife K. O'Connell, Anna Tseng, Nicholas A. Crossland, Gustavo Mostoslavsky, Elke Mühlberger.

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
