## [Decision Letter · Decision Letter 0]

26 Aug 2025

Filovirus Infection Disrupts Epithelial Barrier Function and Ion Transport in Human iPSC-Derived Gut Organoids

PLOS Pathogens

Dear Dr. Mühlberger,

Thank you for submitting your manuscript to PLOS Pathogens. After careful consideration, we feel that it has merit but does not fully meet PLOS Pathogens's publication criteria as it currently stands. Therefore, we invite you to submit a revised version of the manuscript that addresses the points raised during the review process.

Please submit your revised manuscript within 60 days Oct 25 2025 11:59PM. If you will need more time than this to complete your revisions, please reply to this message or contact the journal office at plospathogens@plos.org. Please include the following items when submitting your revised manuscript:

We look forward to receiving your revised manuscript.

Kind regards,

Ronald N. Harty

Guest Editor

PLOS Pathogens

Matthias Schnell

Section Editor

PLOS Pathogens

Editor-in-Chief

PLOS Pathogens

orcid.org/0000-0003-2946-9497

Editor-in-Chief

PLOS Pathogens

orcid.org/0000-0002-7699-2064

**Additional Editor Comments:**

Overall, the Reviewers found the work to be significant and of high quality. However, Reviewer 3 raised a number of major concerns that need to be addressed. Particular attention should be given to comments #1 regarding donor variability, #2 regarding virus quantification, and #3 regarding MOIs used.

**Journal Requirements:**

At this stage, the following Authors/Authors require contributions: Elizabeth Y. Flores, Adam J. Hume, Judith Olejnik, Aditya Mithal, Andrew D’Amico, MengWei Yang, Pushpinder Bawa, Feiya Wang, Aoife K. O’Connell, Anna Tseng, Nicholas A. Crossland, Gustavo Mostoslavsky, and Elke Mühlberger. Please ensure that the full contributions of each author are acknowledged in the "Add/Edit/Remove Authors" section of our submission form.

https://journals.plos.org/plospathogens/s/submission-guidelines#loc-parts-of-a-submission

4) We do not publish any copyright or trademark symbols that usually accompany proprietary names, eg ©,  ®, or TM  (e.g. next to drug or reagent names). Therefore please remove all instances of trademark/copyright symbols throughout the text, including:

- ® on page: 46 and 59.

- TM on pages: 32, 54, 58, and 59.

5) Please upload all main figures as separate Figure files in .tif or .eps format. For more information about how to convert and format your figure files please see our guidelines: 

6) We notice that your supplementary Figures, and Tables are included in the manuscript file. Please remove them and upload them with the file type 'Supporting Information'. Please ensure that each Supporting Information file has a legend listed in the manuscript after the references list.

7) Please ensure that the funders and grant numbers match between the Financial Disclosure field and the Funding Information tab in your submission form. Note that the funders must be provided in the same order in both places as well.

**Reviewers' Comments:**

Reviewer's Responses to Questions

**Part I - Summary**

Reviewer #1: This manuscript represents a significant step forward in the study of host–virus interactions, utilizing organoid systems to bridge the gap between simplified cell culture and complex in vivo models.

Using iPSC-derived intestinal and colonic organoids, the authors characterized Ebola (EBOV) and Marburg (MARV) virus infection in these 3D tissue models. They confirmed expression of intestinal markers (CDX2 and VIL1) and demonstrated that both EBOV and MARV replicated efficiently in colonic organoids (HCOs), with 10–20% of cells positive for viral antigen by day 3 post-inoculation and up to ~40% EBOV-positive cells when quantified by NP-specific flow cytometry.

Bulk RNA-seq analysis of infected organoids revealed transcriptional responses enriched in pathways related to epithelial structure and barrier function, as expected. Notably, MARV infection induced a robust interferon (IFN) response in HCOs, whereas EBOV infection suppressed this pathway, despite both viruses encoding IFN antagonists.

For comparison, the authors examined infection in primary intestinal organoids derived from two donors. Infection patterns were less robust, varied between donors, and were largely restricted to enterocytes. Finally, forskolin-induced swelling assays demonstrated that EBOV and MARV infection disrupted cAMP signaling and impaired organoid functional integrity.

Reviewer #2: This manuscript describes the development of iPS-derived human intestinal and colonic organoid systems for studying Marburg and Ebolavirus pathogenesis in vitro. Protocols are published to do this procedure and the authors show that their organoids express appropriate markers of their cell types of interest. The organoids can be productively infected with EBOV and MARV and evidence of replication and cell spread of virus are shown. RNAseq analysis of cellular transcripts identified altered gene expression patterns. They then examine the effect of viral infection on cellular barrier function using a common forskolin-induced CTFR assay. They observe a cell=type dependent response, wherein infection does not alter intestinal organoids, but disrupts colonic organoid barrier function. Finally, they examined infection of human primary cell organoids, which displayed similar phenotypes as the iPSC-derived organoids. Overall, I found this study to be of high quality and of significance to the field in developing human organoid model systems to study EBOV/MARB infection. My critiques are minor.

Reviewer #3: Flores et al. present three different organoid model to assess molecular aspects of Ebola and Marburg virus pathogenesis in the gastrointestinal tract. They conclude that the here used two iPSC-derived models and the one primary cell-derived model recapitulate the filovirus infections in vivo. They further use bulk RNA sequencing to analyze transcriptomic changes upon infection. They use in situ hybridization and a forskolin-swelling assay to validate findings from their transcriptome analysis. The study encompasses a broad spectrum of analyses, validation of RNA seq data, and the new models significantly add to the field for further pathogenic studies.

**Part II – Major Issues: Key Experiments Required for Acceptance**

Reviewer #1: The authors have developed gastrointestinal organoid models for filovirus infection and employed multiple assays to characterize host responses. While they present transcriptomic data alongside in situ hybridization data, it remains unclear whether these transcriptional changes translate into secretion of cytokines, chemokines, or interferons. Can ELISA or similar assays detect these secreted proteins in the gastrointestinal organoid system?

Reviewer #2: (No Response)

Reviewer #3: 1. The authors appear to use only one donor for their iPSC-derived organoids and only two for their primary cell-derived organoids. Given donor-to-donor variability, which is a known issue with organoids and primary cultures, biases cannot be excluded. Therefore, it is absolutely needed to verify all results in at least 3 different donors.

2. Even though an increase in antigen positive cells between 1 dpi and 3 dpi in Fig 3 and 6 is visible and an increase in viral transcripts in Fig 5, a successful dissemination of infectious virus cannot be fully concluded from the data. Quantification of disseminated virus (infectious titers) in the supernatant of the organoids is missing and absolutely needed to validate the statement of successful viral dissemination.

3. It is unclear and not mentioned in the M&M section how the MOI was calculated to infect the organoids, since all organoids have a different number of cells. Furthermore, an MOI of 10 seems to be rather high compared to standard 2D cell culture experiments. The authors need to justify the physiological relevance of using such a high MOI and discuss why a lower MOI was not an option?

4. In Figure 3 D the infection rate of EBOV infected HCOs was quantified using flow cytometry, but a similar quantification is lacking for MARV-infected HCOs and for the HIOs. Why? The authors should add a quantification to assess infection rate, especially since quantification of virus in the supernatant or in “lysed” organoids is lacking.

5. Figure 3 I, K: How was the immune-reactive tissue quantified?

6. For the determination of differentially expressed genes upon infection “mock- infected control” were used. However, it is unclear what these are exactly. Since residues, like signaling molecules, cytokines, chemokines etc., from the virus propagation and purification can affect sensitive experiments like these unrelated to the actual viral infection, conditioned media (e.g UV-light/ irradiation virus inactivated media) should ideally be used. The authors should elucidate and ideally discuss limitations of their mock control if conditioned media was not used.

7. What was the minimal RPKM cut off value for the analysis of DEG?

8. The statement that EBOV and MARV infection leads to extensive disruption of epithelial structure is overinterpreting the RNA seq data. In order to support this conclusion, this needs to be verified on a protein/phenotype basis. How does the epithelial layer look like in IFA of infected organoids?

9. Why does the expression of gut markers increase in the 3 dpi mock HCOs? Shouldn’t they stay consistent in the mock infection?

10. The authors find a greater epithelial expression of innate immune markers upon MARV infection compared to EBOV infection. Why is that?

11. The authors use FISH to quantify mRNA expression of innate immune markers in the organoids wanting to validate findings from bulk RNA seq. However, a quantification based on protein expression instead of mRNA, especially in the context of imaging tissues, is a much better readout and validation and needs to be added to validate the claims. Additionally, the description of Fig. S3 is inconclusive showing one point, saying two independent infections (n=3). How many images were analyzed to rule out using one “good-looking” image?

12. the entire manuscript is very wordy and needs ediitng for a reader to stay engaged. It is very hard to read.

**Part III – Minor Issues: Editorial and Data Presentation Modifications**

Reviewer #1: Figure 3J: the MARV 3 dpi panel reads EBOV

Figure 8D: Any potential statistics?

Reviewer #2: 1. In Fig. 5, I was surprised that significant gene expression changes were observed in mock organoids from day 1 to 3, particularly in the gut markers which are highly upregulated. This suggests that the organoids were incompletely differentiated when they were initially infected. While I don’t expect you to redo the study with fully differentiated iPSC- derived organoids. You should at least discuss this result.

2. In Fig. 7, is there a reason why only one time pint is used? It is not clear whether this normalized data, wherein one data point is normalized to mock reflects a decrease in reads in mock cells or an increase in reads in infected cells (see point 1). This caveat should be discussed.

3. Fig. 8C needs statistical significance.

Reviewer #3: 1. l. 52: Is the use of both comma and em dash here correct? Please double check.

2. l. 58: “provide insights into the transmission” and pathogenesis?

3. Figure 1: Brightfield images in D are very difficult to see due to the small size. Can the size be increased, or a smaller window used so individual cells are easier to see?

4. l. 121: please first introduce dpi upon first usage in the manuscript.

5. Figure 2: 7 dpi images for HIOs are missing?

6. l. 207: “1- and 3-dpi” should be without hyphenation since it has been written without it in the rest of the manuscript.

7. Figure 4 G-H: to increase readability consider erasing the “Hallmark” in front of every hallmark name. These graphs are very small. Consider only displaying the significant once in the main figure to further increase readability of these types of figures in general.

8. Figure 5: Not readable and hard to draw conclusion from in its current state. Consider highlighting certain genes and only showing those and breaking them up into groups while showing an even bigger image of the full set in the supplement.

9. ll. 862-863; ll. 898-899; ll. 957-958: safe spaces between number and unit to not separate them in between two lines.

10. l. 878; l. 887: “1X PBS” should be “1× PBS”.

11. l. 929 : Is the version number missing for MiniNet or is the line intentional?

12. l. 942f. “1- and 3- days dpi” should be “1 and 3 dpi”.

13. l. 951: “10-minute” should be “10 minute”.

14. l. 989: Throughout the M&M section spaces between number, times and g was written (200 × g), but here no space. Please decide on one across the entire section.

15. Table 1: The title of table one is inaccurate as the table not only lists antibodies but antibodies, dyes, and reagents. Please correct.

16. The table design of table 3 is a bit unfortunate since, when printed does not show the lines making it hard to decipher and read. A lot of the words and names are split in weird ways making it harder to read. Think about switching this table into landscape to make it readable.

PLOS authors have the option to publish the peer review history of their article (what does this mean? ). If published, this will include your full peer review and any attached files.

**Do you want your identity to be public for this peer review?** For information about this choice, including consent withdrawal, please see our Privacy Policy .

Reviewer #1: No

Reviewer #2: No

Reviewer #3: No

**Figure resubmission:**

**Reproducibility:**



---

## [Editor Report · Decision Letter 1]

15 Oct 2025

PPATHOGENS-D-25-01734R1

Filovirus Infection Disrupts Epithelial Barrier Function and Ion Transport in Human iPSC-Derived Gut Organoids

PLOS Pathogens

Dear Dr. Mühlberger,

Thank you for submitting your manuscript to PLOS Pathogens. After careful consideration, we feel that it has merit but does not fully meet PLOS Pathogens's publication criteria as it currently stands. Therefore, we invite you to submit a revised version of the manuscript that addresses the points raised during the review process.

Please submit your revised manuscript within 30 days Dec 14 2025 11:59PM. If you will need more time than this to complete your revisions, please reply to this message or contact the journal office at plospathogens@plos.org. Please include the following items when submitting your revised manuscript:

We look forward to receiving your revised manuscript.

Kind regards,

Matthias Johannes Schnell, PhD

Section Editor

PLOS Pathogens

Matthias Schnell

Section Editor

PLOS Pathogens

Sumita Bhaduri-McIntosh

Editor-in-Chief

PLOS Pathogens

orcid.org/0000-0003-2946-9497

Michael Malim

Editor-in-Chief

PLOS Pathogens

orcid.org/0000-0002-7699-2064

**Reviewers' Comments:**

**Figure resubmission:**
---

## [Editor Report · Decision Letter 2]

6 Nov 2025

Dear Dr. Mühlberger,

We are pleased to inform you that your manuscript 'Filovirus Infection Disrupts Epithelial Barrier Function and Ion Transport in Human iPSC-Derived Gut Organoids' has been provisionally accepted for publication in PLOS Pathogens.

Best regards,

Matthias Johannes Schnell, PhD

Section Editor

PLOS Pathogens

Matthias Schnell

Section Editor

PLOS Pathogens

Sumita Bhaduri-McIntosh

Editor-in-Chief

PLOS Pathogens

orcid.org/0000-0003-2946-9497

Michael Malim

Editor-in-Chief

PLOS Pathogens

orcid.org/0000-0002-7699-2064
---

## [Editor Report · Acceptance letter]

Dear Dr. Mühlberger,

We are delighted to inform you that your manuscript, "Filovirus Infection Disrupts Epithelial Barrier Function and Ion Transport in Human iPSC-Derived Gut Organoids," has been formally accepted for publication in PLOS Pathogens.

Best regards,

Sumita Bhaduri-McIntosh

Editor-in-Chief

PLOS Pathogens

orcid.org/0000-0003-2946-9497

Michael Malim

Editor-in-Chief

PLOS Pathogens

orcid.org/0000-0002-7699-2064